# Different population dynamics in the supplementary motor area and motor cortex during reaching

A.H. Lara[1], J.P. Cunningham [2,3,4,5] & M.M. Churchland[1,3,4,6]

Neural populations perform computations through their collective activity. Different computations likely require different population-level dynamics. We leverage this assumption to examine neural responses recorded from the supplementary motor area (SMA) and motor cortex. During visually guided reaching, the respective roles of these areas remain unclear; neurons in both areas exhibit preparation-related activity and complex patterns of movement-related activity. To explore population dynamics, we employ a novel "hypothesis-guided" dimensionality reduction approach. This approach reveals commonalities but also stark differences: linear population dynamics, dominated by rotations, are prominent in motor cortex but largely absent in SMA. In motor cortex, the observed dynamics produce patterns resembling muscle activity. Conversely, the non-rotational patterns in SMA co-vary with cues regarding when movement should be initiated. Thus, while SMA and motor cortex display superficially similar single-neuron responses during visually guided reaching, their different population dynamics indicate they are likely performing quite different computations.

[1] Department of Neuroscience, Columbia University Medical Center, New York, NY 10032, USA. [2] Department of Statistics, Columbia University, New York, NY 10027, USA. [3] Zuckerman Mind Brain Behavior Institute, Columbia University, New York, NY 10027, USA. [4] Grossman Center for the Statistics of Mind, Columbia University, New York, NY 10027, USA. [5] Center for Theoretical Neuroscience, Columbia University Medical Center, New York, NY 10032, USA. [6] Kavli Institute for Brain Science, Columbia University Medical Center, New York, NY 10032, USA. Correspondence and requests for materials should be addressed to M.M.C. (email: mc3502@columbia.edu)

A basic goal of motor physiology is to characterize cortical responses during voluntary movement. Characterization requires identifying parameters with which neural activity covaries: e.g., muscle activation, hand velocity, or higher-order features[1–3]. Yet neural activity contains structure beyond that reflecting movement parameters[4–10], including structure reflecting dynamics that generate and control movement[6,7,11–14]. Here we consider the supplementary motor area (SMA) and motor cortex, and ask whether they obey similar or different population-level dynamics. It is hypothesized that SMA is specialized for sequences and/or internally initiated movements[15–25]. SMA is nevertheless active during single movements[26], including externally prompted movements[27–32]. The relative roles of SMA and motor cortex during non-sequential movements thus remain uncertain (for review see ref. [17]). There exist at least three possibilities. First, SMA and motor cortex may perform redundant computations. By analogy, the frontal eye fields and superior colliculus have different specializations, but respond similarly during simple saccades[33]. Second, SMA and motor cortex may process different kinds of information, yet still perform computations subserved by similar dynamics. This could be consistent with the proposal that different cortical areas perform a canonical operation on different inputs[34,35]. Third, SMA and motor cortex may perform very different computations via different classes of dynamics.

Known differences between SMA and motor cortex do not resolve these possibilities. Anatomy suggests parallel complementary contributions[36], but that is consistent with either different or similar computations. In support of a higher-order role for SMA, SMA facilitates muscle activity less strongly[37], is less responsive to proprioceptive input[26,38], is more responsive during ipsilateral arm movements[26], displays signals reflecting movement outcome[39], and contributes to learning between movements[40]. Yet during standard visually guided movements, aspects of SMA and motor cortex responses are extremely similar, suggesting that they "operate in parallel"[21] and may make largely redundant contributions.

We addressed the relative contributions of SMA and motor cortex by examining the population-level dynamics that presumably subserve network computations. This comparison is aided by recent characterizations of motor cortex dynamics. During reaching, the motor cortex population displays a "central motif" composed of two aspects: a condition-invariant shift in state[5] immediately followed by state trajectories following rotational dynamics[6,12,41]. This same motif is naturally displayed by network models trained to generate muscle activity patterns[42]. For such models, the central motif reflects the underlying computation. The condition-invariant shift initiates movement by bringing the state to a region where rotational dynamics dominate. The resulting oscillatory patterns form a basis for multiphasic muscle commands. Rotational dynamics, related to quasi-periodic sub-movements, have also been observed in LFP[43]. Given the proposed connections between dynamics and function, the central motif represents a natural point of comparison.

We recorded neural responses from SMA and motor cortex while monkeys executed reaches to radially arranged targets. When examined using the population vector and population PSTH, SMA and motor cortex showed similar structure, including preparatory and movement-related activity that covaried with reach direction. To examine dynamics, we employed a novel "hypothesis-guided" dimensionality reduction (HDR) approach that translates a hypothesis into a cost function. Our cost function sought projections where some dimensions capture a condition-invariant shift in state while other dimensions capture trajectories described by generic linear dynamics.

Both SMA and motor cortex displayed a large, similarly organized, condition-invariant shift in population state just before movement initiation. Thus, the first aspect of the central motif was almost perfectly shared between the two areas, possibly constituting a shared signal related to movement initiation[5]. Yet in terms of dynamics, SMA and motor cortex were quite different. SMA activity was not well described by linear dynamics, displayed weak rotational structure overall, and lacked the 1.5–3 Hz rotations previously reported in motor cortex. In contrast, HDR identified multiple dimensions where motor cortex activity obeyed approximately linear dynamics. Motor cortex dynamics were dominated by rotations, even though HDR did not specifically seek rotations. Rotations occurred in the 1.5–3 Hz range and produced response features matching multiphasic aspects of muscle activity. Although SMA lacked the clear dynamical structure found in motor cortex, it contained a complementary type of information: SMA activity co-varied with the "higher-level" task constraints that determined when movement could be initiated.

In summary, only in SMA did activity vary strongly with higher-level task requirements. SMA and motor cortex both shared a large signal previously shown to be temporally locked to movement initiation. Finally, only motor cortex showed strong rotational dynamics. These different dynamics, and the different types of information carried by neural activity, argue that SMA and motor cortex are performing very different computations.

## Results

**Task**. Two monkeys (Ba and Ax) executed radial reaches in eight directions across three contexts: cue-initiated, self-initiated, and quasi-automatic. These contexts differed in the task requirements governing how and when movement should be initiated. The cue-initiated context employed the standard instructed-delay paradigm: a randomized delay period (0–1000 ms) separated target onset from an explicit go cue. In the self-initiated context, monkeys were free to reach upon target presentation, but waiting longer yielded larger rewards up to a limit at 1200 ms. The quasi-automatic context was similar to the cue-initiated context, but the go cue was the onset of target motion along a radial path toward the screen's edge. This context evoked low-latency reaches that intercepted the target mid-flight. Target and central touch-point color (red, blue, or yellow) cued the context. Trials were interleaved. Reaches had similar trajectories across contexts (Fig. 1a) but tended to be slightly faster for the quasi-automatic context (Fig. 1b, yellow).

In a separate study[44], we exploit these contexts to examine preparatory neural events in motor cortex. In the present study, we compare movement-related dynamics between areas. Given this goal, the advantage of different contexts is that they elicit responses across a greater range of situations—including situations that may differentially engage SMA. We analyzed only trials with sufficient time, between target and movement onset, for clear preparatory activity to be established. This allowed analysis to concentrate on movement-related dynamics. For cue-initiated and quasi-automatic contexts, we analyzed trials with delays >400 ms. For the self-initiated context, the monkeys' behavior provided the desired separation.

**Neural and muscle recordings**. We recorded neural responses from SMA (141 and 186 neurons for monkey Ba and Ax) and motor cortex (129 and 172 neurons). Recordings used single electrodes or 24-channel linear electrode arrays. Recordings were made from regions where electrical stimulation produced arm movements. Figure 1c illustrates where electrode penetrations

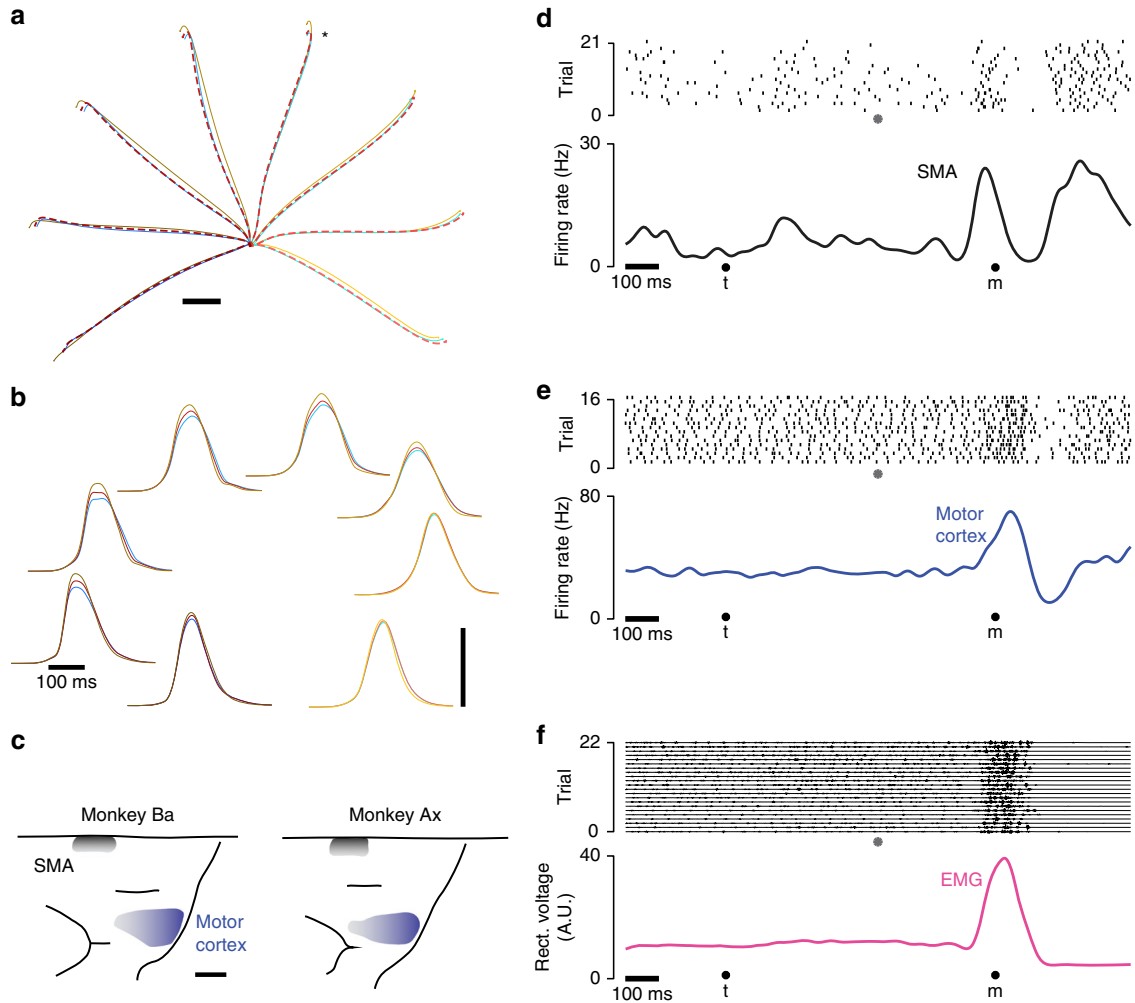

**Fig. 1** Illustration of behavior and physiological recordings. **a** Average reach trajectories for the eight directions and three contexts. Red, blue and yellow traces show average hand position during curing cue-initiated, self-initiated, and quasi-automatic reaches. Traces are shaded from dark to light based on reach direction. Red traces are dashed to allow visualization of blue traces with which they often overlap. Data shown are for monkey Ba, and were similar for monkey Ax. Star indicates the reach direction for which neural/muscle data are shown in panel (**d**). Scale bar shows 2 cm. **b** Average hand-velocity profiles corresponding to the trajectories in panel (**a**). Scale bar shows 1 m/s. **c** Reconstructions of surface landmarks based on MRIs (see Supplementary Figure 1 for example MRI sections). Shaded regions indicate where penetrations entered cortex, and are shaded darker to indicate where recordings included deeper locations. Scale bar shows 5 mm. **d** Raster plot of spikes recorded from one SMA neuron for one reach direction during the quasi-automatic condition. Data in this and subsequent panels are for monkey Ax. Data to the left of the gray symbol are aligned to target onset (t) and data to the right are aligned to movement onset (m). Filtering and averaging of spike-trains yields a smooth firing rate versus time (black trace) that interpolates across the concatenation at the time indicated by the gray symbol. Filtering used a narrow (20 ms) Gaussian to ensure high-frequency aspects of the response were not lost. **e** Same as in **d** but for a neuron recorded from motor cortex. **f** Similar to **d** and **e** but for EMG recorded from the medial deltoid. Voltage traces show a mixture of discrete and (especially during movement) overlapping events. Discontinuities resulting from concatenation of target-locked and movement-locked data are small and barely visible. Rectification, filtering and averaging produces a continuous trace summarizing average EMG intensity (magenta trace)

entered (Supplementary Fig. 1 shows structural MRI). For SMA, nearly all recordings were made relatively deep, from the medial wall. For motor cortex, we recorded from sulcal primary motor cortex, surface primary motor cortex and the immediately adjacent aspect of caudal PMd. Across these recordings we observed the expected gradient of stronger preparatory activity on the surface versus sulcus. Yet this tendency was far from complete and responses formed a continuum with no noticeable discontinuity with location. We thus analyzed primary and premotor cortex recordings together as a single motor cortex population. This is consistent with our prior finding that primary and premotor cortex display similar dynamics when analyzed separately or together[6].

Spike-trains during a target-locked epoch and a movement-locked epoch were concatenated (Fig. 1d, e; gray circles indicate concatenation time) allowing computation of an across-trial firing rate that spans both events with a representative separation (traces at bottom of each panel). This unified rate was useful for visualizing preparatory and movement-related events together. However, all analyses of dynamics focus on the movement-aligned epoch, after the time of concatenation. Muscle activity was recorded percutaneously from the major muscles of the upper arm (13 and 10 recordings for monkey Ba and Ax). We employed the standard technique of rectifying the voltage traces, which provides a net measure of the activity of many motor units. The average response of a given muscle for a given condition

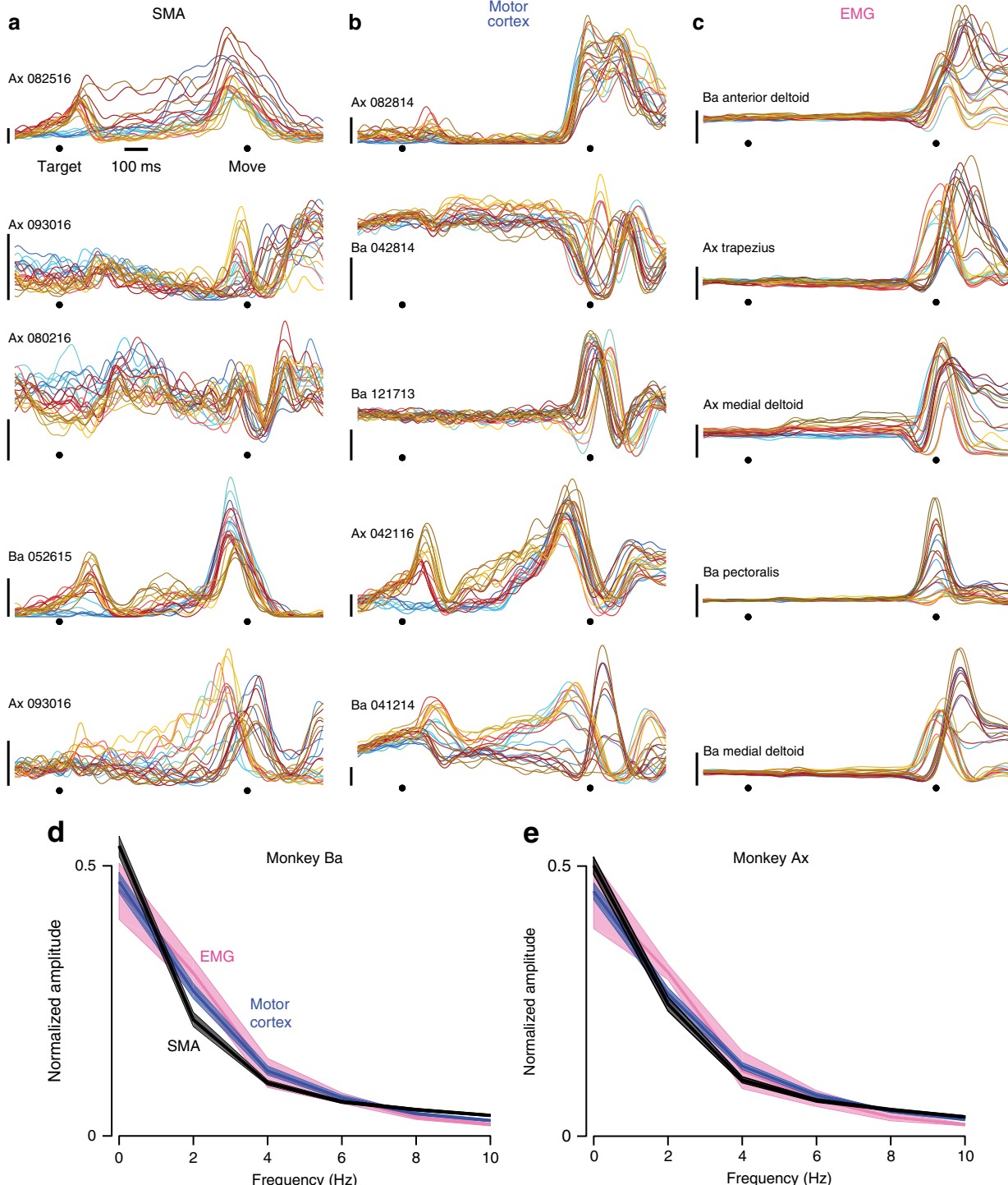

**Fig. 2** Example responses of single neurons and muscles. **a** Example responses of five neurons recorded from SMA. Each colored trace plots the trial-averaged firing rate for one condition, computed as illustrated in Fig. 1. Red, blue, and yellow traces correspond to the three contexts. Darker/lighter traces are for reaches to the left/right. (Same color scheme as in Fig. 1a, b). Scale bars indicate 20 spikes/s. **b** Same as **a**, but for five neurons recorded from motor cortex. **c** Same as **a** and **b**, but for five example muscle recordings. The bottom of the vertical scale bars indicates zero EMG activity, but the scale is otherwise arbitrary. **d** Frequency spectrum, computed via the Fourier transform, for the three populations. Frequency content was computed per neuron/muscle, over the temporal interval from −250 to 250 ms relative to movement onset. Frequency content was then normalized and averaged. Envelopes show 95% confidence intervals computed via bootstrap, resampling neurons/muscles. Data are for monkey Ba. **e** Same as **d**, but for monkey Ax

(Fig. 1f, magenta trace) was then computed just as for the neurons.

On average, firing rates were higher in motor cortex: peak firing rate averaged 75 and 77 spikes/s (monkey Ba and Ax) versus 43 and 53 spikes/s for SMA. Individual-neuron firing rate

estimates thus tended to be slightly noisier for SMA. Otherwise, single-neuron responses were in many ways similar (Fig. 2a, b). For both areas, firing rates varied with reach direction before and during movement (darker/lighter traces correspond to leftwards/rightwards reaches). During movement, responses often exhibited

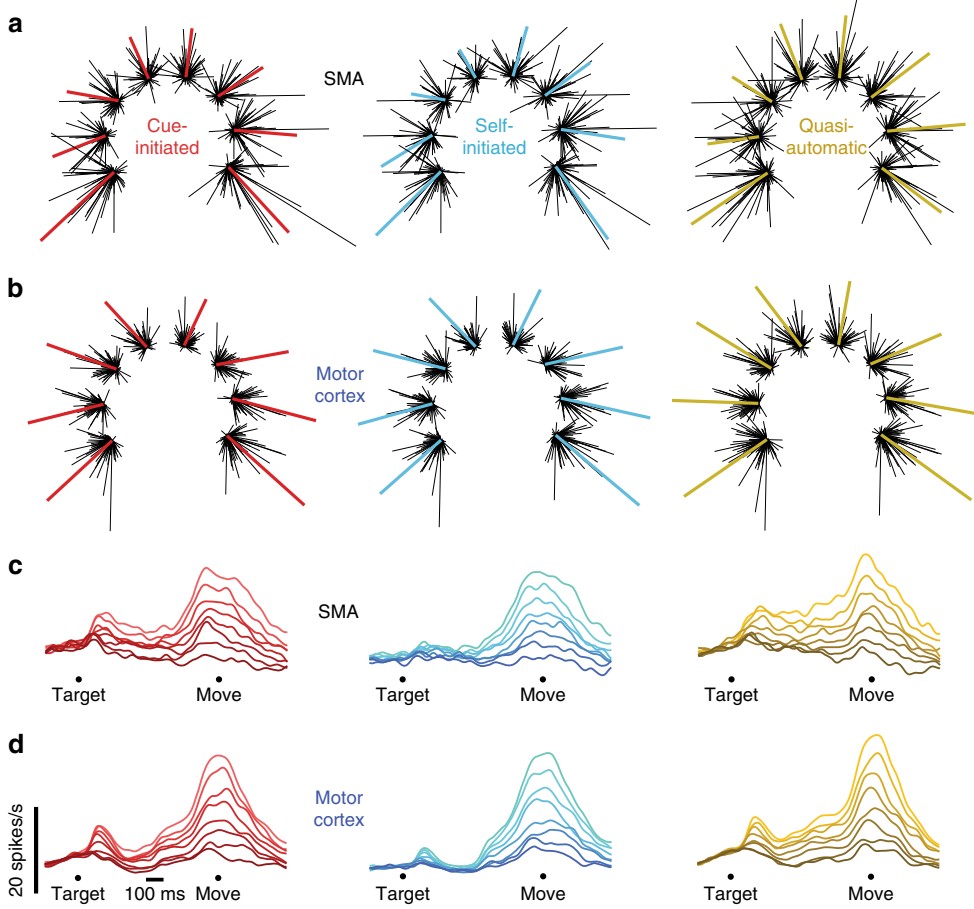

**Fig. 3** Population vectors and population PSTHs for monkey Ba. **a** SMA population vectors for each of the three contexts. Each cluster of lines corresponds to one of the eight reach directions. The length of each black line indicates the firing rate of one neuron for that direction and context, and points in that neuron's preferred direction. Colored traces plot (on a different scale) the sum of those vectors. For each neuron, the preferred direction was computed based on firing rates across all directions and contexts. **b** Motor cortex population vectors. Scaling is arbitrary and differs between the two brain areas, as they had different average firing rates (see main text). **c** SMA population PSTHs for the three contexts. Shading is ordered from most-preferred direction (lightest) to least-preferred (darkest) within each context. **d** Motor cortex population PSTHs

multiple peaks and/or phases that varied with condition. Individual neurons displayed a variety of diverse and complex responses[45]. Directional selectivity often differed before and during movement, or early versus late during movement[45–50]. Robust activity was typically present across contexts. The above results are consistent with the finding that SMA has directional responses not unlike those in motor cortex, and is responsive during both self-initiated and externally cued movements[27,28,31].

Muscle responses were also often complex and multiphasic (Fig. 2c). Muscles did not show changes in activity until just before movement onset, with some exceptions. For example, baseline activity of the medial deltoid for monkey Ax (third subpanel) increased slightly overall, well before movement (more so for the quasi-automatic context). These changes presumably reflect a slight tensing in anticipation of the pending reach. Relative to neural activity in the same time-range, changes in muscle activity were sparse, small, and weakly directional.

**Standard population analyses for SMA and motor cortex.** The population vector[51], a summary based on directional aspects of activity, behaved similarly for SMA and motor cortex (Fig. 3a, b, monkey Ba; Supplementary Fig. 2, monkey Ax). We found only small differences between the two areas. We first considered the vector magnitude, relative to the firing rates of the contributing

neurons. Comparing SMA with motor cortex, vector magnitude was similar: slightly larger for monkey Ba (5%, N.S.) and smaller for monkey Ax (24%, $p < 0.01$). For both monkeys, the SMA population vector was slightly, but not significantly, less-well aligned with target direction.

We also employed the population PSTH: for each neuron, we identified the most-preferred condition (the condition that evoked the largest firing rate over a 500 ms window centered on movement onset) then averaged that response across neurons. A similar average was produced for the least-preferred condition and all intermediate conditions. This analysis was repeated separately for each context. The resulting population PSTHs (Fig. 3c, d) reveal that preparatory tuning (variation of firing rate with target direction) developed shortly after target onset for the cue-initiated and quasi-automatic contexts (red and yellow) and somewhat later for the self-initiated context (blue). Stronger movement-related tuning then arose ~150 ms before movement onset.

Population PSTHs showed only modest differences between areas. Population PSTHs reveal slightly stronger preparatory versus movement-period tuning for SMA. This was consistent with single-neuron observations; the median ratio of preparatory to movement tuning was higher for SMA versus motor cortex (0.79 versus 0.52 for monkey Ba; 0.81 versus 0.64 for monkey Ax; $p < 0.01$ for both via rank sum test). For monkey Ba tuning tended

to remain high slightly longer for SMA, but the opposite tendency was observed for monkey Ax. Despite these modest differences, population PSTHs were strongly correlated between areas. For monkey Ba, correlations were 0.92, 0.96, and 0.92 for the three contexts. For monkey Ax, correlations were 0.94, 0.96, and 0.92. The lower bound of the 95% confidence intervals was >0.90 for all comparisons.

Frequency spectra also revealed only modest differences between SMA and motor cortex (Fig. 2d, e). There was slightly greater power in the 1.5–3 Hz range for motor cortex (and for the muscles) versus SMA. This range includes frequencies followed by rotational dynamics in prior studies. This is potentially suggestive, but no more: SMA also had considerable power in this range.

Thus, SMA and motor cortex appear reasonably similar when analyzed via these standard methods. However, these methods are suited to revealing certain aspects of the population response but not others. For example, the frequency spectrum does not distinguish between power organized into coherent rotations versus disorganized trajectories. The population vector confirms the presence of directional responses but does not capture other response properties. Population PSTHs reveal the rough envelope of tuning, but not how activity evolves within that envelope. Indeed, population PSTHs incorrectly suggests that tuning remains consistent with time. Analyses of dynamics provide an alternative approach that might sidestep these limitations.

**Hypothesis-guided dimensionality reduction**. We employ the perspective that single-neuron responses reflect latent variables that are shared across the population[11,52–57] and can be estimated using dimensionality reduction methods[58–63]. A standard linear model of each neuron's response is:

$$a_{t,n} = \sum_k x_{t,k} w_{k,n} \qquad (1)$$

where $a_{t,n}$ is the firing rate of neuron $n$ at time $t$, $x_{t,k}$ is the value of the $k$th latent variable, and $w_{k,n}$ determines the contribution of that latent variable to the response of neuron $n$. A key question is whether the evolution of $\mathbf{x}_t$, the vector of latent variables, is described by a dynamical flow field. Is it the case that $\dot{\mathbf{x}}_t \approx f(\mathbf{x}_t)$ for some function $f(\cdot)$, perhaps with some linear approximation: $\dot{\mathbf{x}}_t \approx \mathbf{x}_t D$? If so, is $D$ dominated by rotations or other forms of dynamics? These questions cannot be addressed at the single-neuron level. Neither multiphasic firing rates[6] nor data smoothness imply rotational dynamics[64]; much depends on how phases are coordinated across neurons and conditions[65].

We previously examined motor cortex responses using a method, jPCA, that seeks latent variables described by rotational dynamics. jPCA has two shortcomings given our present goals. First, when comparing areas, we wish to make fewer assumptions regarding the form of dynamics. Second, the central motif predicted by motor-cortex network models includes both rotational dynamics and a condition-invariant shift of the neural state[42]. We previously resorted to multiple methods (dPCA[66] followed by jPCA) to test for the presence of the central motif[5]. That approach is suboptimal; latent variables should ideally be found in a unified fashion.

To do so, we employ a hypothesis-guided dimensionality reduction (HDR) methodology. Recent work observes that most dimensionality reduction methods implicitly or explicitly employ a cost function[67], and that different cost functions embody different hypotheses. For example, PCA embodies the simple hypothesis that the most relevant signals are the largest signals, while dPCA[66] posits that different dimensions contain activity that co-varies with different task parameters. Here we follow this

lead and leverage the suggestion that "future linear dimensionality reduction algorithms can be derived in a simpler and more principled fashion"[67]. We adopt a cost function tailored to our specific hypothesis: there may exist a projection of the data that captures a large percentage of response variance, with some dimensions capturing condition-invariant structure and other dimensions capturing structure described by linear dynamics.

We consider the data matrix, $A$, where each column contains the firing rate of one neuron across times and conditions. We estimate the latent variables as a projection, $X = A W^T$, where each column of $X$ contains the values of one latent variable across times and conditions. The rows of the orthogonal matrix $W$ are the "neural dimensions", found by minimizing a cost function $f(W)$. Because the above hypothesis contains three components, we employ a tripartite cost function:

$$f(W) = f_{\text{rec}}(W) + f_{\text{invar}}(W) + f_{\text{dyn}}(W) \qquad (2)$$

The first term, $f_{\text{rec}}(W)$, is identical to the PCA cost function, and is small if the latent variables capture considerable variance (i.e., if the firing rates in $A$ are accurately reconstructed by $A_{\text{rec}} = XW$). The second two terms relate to the hypothesis that there exists condition-invariant structure in some dimensions and dynamical structure in other, orthogonal dimensions. If so, an appropriate $W$ will result in latent variables $X = [X_{\text{invar}}, X_{\text{dyn}}]$. $X_{\text{invar}}$ are latent variables that vary with time but not condition. $X_{\text{dyn}}$ are latent variables whose evolution obeys linear dynamics. $f_{\text{invar}}(W)$ is small if $X_{\text{invar}}$ varies strongly with time but not condition. $f_{\text{dyn}}(W)$ is small if the fit $\dot{X}_{\text{dyn}} \approx X_{\text{dyn}} D$ is good for some choice of $D$. Equations for $f_{\text{rec}}(W)$, $f_{\text{invar}}(W)$, and $f_{\text{dyn}}(W)$ are provided in the Methods. By minimizing $f(W)$ we ask the question: does there exist a projection of the population response that captures considerable variance and has the hypothesized condition-invariant and dynamical structure?

For each population, we minimized $f(W)$ via gradient descent on $W$, then used those dimensions to find the latent variables. We refer to $X_{\text{invar}}$ as the projection onto "condition-invariant dimensions". Of course, whether $X_{\text{invar}}$ actually displays the hypothesized condition-invariant structure is an empirical question. We refer to $X_{\text{dyn}}$ as the projection onto "dynamical dimensions". Again, whether $X_{\text{dyn}}$ actually displays dynamical structure is an empirical question. We set the total number of dimensions to six, and sought two condition-invariant dimensions and four dynamical dimensions.

As with other methods, it is important to know whether dimensions capture considerable data variance. Notably, PCA minimizes $f_{\text{rec}}(W)$ while we are asking HDR to minimize $f_{\text{rec}}(W) + f_{\text{invar}}(W) + f_{\text{dyn}}(W)$. Thus, the variance captured by PCA is the maximum that could possibly be captured by HDR; HDR must sacrifice some captured variance as it seeks the hypothesized structure. In practice, the six HDR dimensions captured only modestly less variance than the first six PCs, and at least as much variance as PCs 2-7. This was true for both monkeys, both cortical areas, and the muscle populations (Supplementary Fig. 3). Employing more than six dimensions captured slightly more variance but yielded little improvement in capturing condition-invariant or dynamical structure.

HDR optimizes jointly for all aspects of the hypothesized structure. In contrast jPCA employs PCA or dPCA and then seeks rotational structure[5,6], which could cause structure to be missed. Unlike jPCA, the present use of HDR does not focus on rotations per se, reducing concerns that the method imposes a particular form of dynamics. HDR is thus simultaneously more principled, more powerful, and more conservative that past approaches.

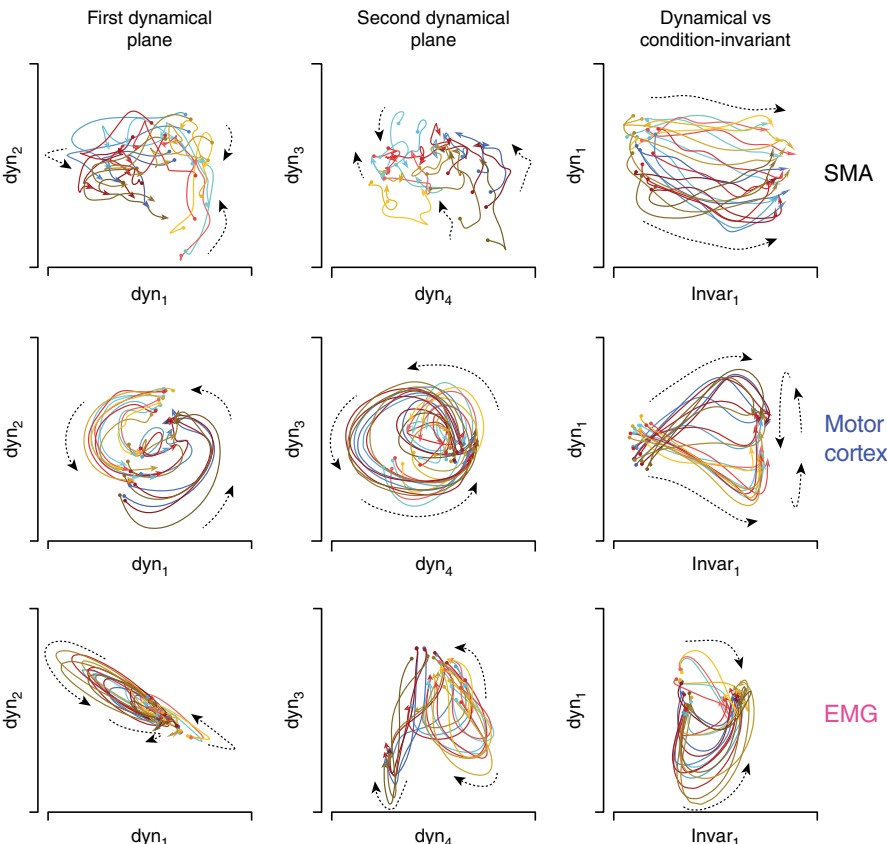

**Fig. 4** State-space plots of the latent variables identified by HDR for monkey Ba. HDR was applied separately for SMA, motor cortex, and muscle populations. Each sub-panel plots the joint evolution of two latent variables (equivalent to a projection of the full neural state onto two dimensions). Each colored trajectory corresponds to one condition and plots the values of the latent variables from −80 to 150 ms relative to movement onset. The beginning and end of each trajectory is indicated by a circle and arrow, respectively. Color-coding is as in Figs. 1a, b and 2a–c: red, blue, and yellow trajectories correspond to the three contexts, while darker/lighter traces correspond to leftward/rightward movements. Dashed lines are included to aid visualization and indicate the local direction of flow (this was often less consistent for SMA and the muscles than for motor cortex). Scaling is arbitrary but the same scale is always used for the horizontal and vertical axis within each subpanel

**Qualitative assessment of dynamical structure**. We plotted pairs of latent variables against one another, yielding projections of the data onto two-dimensional planes. Figure 4 shows three such planes for monkey Ba. We first focus on two "dynamical planes", each showing the projection onto two of four dynamical dimensions. Within a projection, each colored state trajectory describes how those latent variables evolved over time for one condition (one direction/context). For motor cortex (Fig. 4, middle row) both dynamical planes captured rotational structure. Trajectories rotated in the same direction, at approximately the same angular velocity, with phase and amplitude varying across conditions. Trajectories are shown from 80 ms before until 150 ms after movement onset (reaches lasted ~200 ms). Trajectories continued rotating for 50–100 ms after the interval shown (the shorter plotted interval minimizes overlapping traces).

The SMA population (Fig. 4, top row) did not exhibit structure that followed a clear dynamical flow-field (quantification to follow). Rotational structure was weak, and trajectories appeared somewhat disorganized. We stress that this does not imply that the SMA population response is truly disorganized, simply that it is not well described by the hypothesis of approximately linear dynamics. As will be described subsequently, SMA did exhibit other clear structure that could be identified by the HDR approach.

Individual-muscle responses were often multi-phasic, and in many ways resembled individual-neuron responses. Yet the

muscle population showed little coherent rotational tendency (Fig. 4, bottom row). State trajectories often formed loops, but the sign and degree of curvature was inconsistent across conditions. Still, visual inspection suggests some potential commonality between muscle and motor-cortex populations. For example, the first dynamical plane for the muscles resembled that for motor cortex, but viewed from the side. We will return below to this potential connection.

Results were similar for monkey Ax (Fig. 5). The motor cortex population showed robust rotations in two dynamical planes. Some weak rotational structure was present for SMA (more so than for monkey Ba) but the flow-field was not well organized: some trajectories clearly rotated, but many others did not. The muscle population also showed little dynamical structure; looping trajectories were not organized into coherent rotations.

**Quantification of dynamical structure**. For each population, we asked how well $X_{dyn}$, the state in the dynamical dimensions, obeys linear dynamics. We fit with $\dot{X}_{dyn} \approx X_{dyn}D$, where $\dot{X}_{dyn}$ is the time derivative of $X_{dyn}$, and $D$ is the matrix that provides the best fit. $D$ is unconstrained and can capture rotational or other linear dynamics. Results are summarized in Fig. 6a, b (bars labeled "$D$"). For SMA, the dynamical fit was poor ($R^2 = 0.22$, monkey Ba) or moderate ($R^2 = 0.52$, monkey Ax). The dynamical fit was also moderate for the muscle populations: $R^2 = 0.41$ and

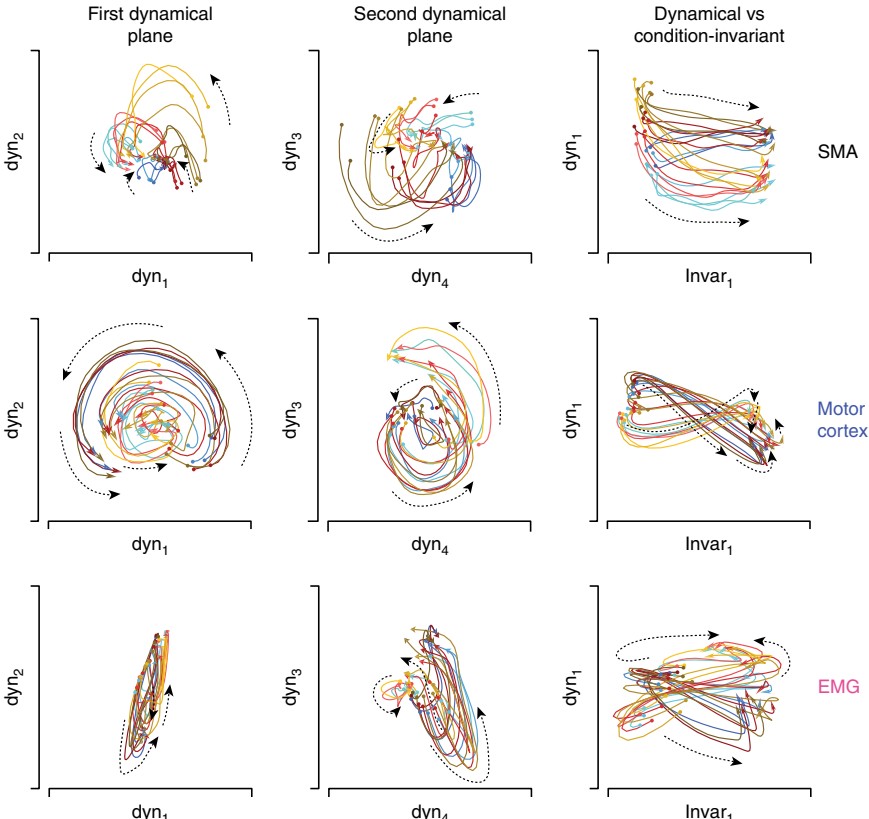

**Fig. 5** State-space plots for monkey Ax. Format as in Fig. 4. Scaling is arbitrary but the same scale is always used for the horizontal and vertical axis within each subpanel. Scaling is sometimes altered across subpanels. In particular, for motor cortex, scaling was reduced for the third column to allow plotting of the first condition-independent latent variable, which had a very large magnitude

0.46. The dynamical fit was best for the motor cortex populations: $R^2 = 0.84$ and 0.76. The difference between SMA and motor cortex was significant ($p < 0.001$ for both monkeys) based on a bootstrap that redrew conditions to establish confidence intervals. We also employed a highly conservative bootstrap that redrew dimensions rather than conditions (Methods). This approach treats the combined SMA and motor cortex latent variables as a large undifferentiated set of latent variables in a common population. The bootstrap then asks how often two arbitrary "areas", each containing a subset of those variables, differ by as much as the empirical SMA and motor cortex datasets. Even via this very conservative method, the difference between SMA and motor cortex was significant: $p < 0.0001$ for monkey Ba and $p < 0.04$ for monkey Ax.

The projections in Figs. 4 and 5 suggest that a large difference between SMA and motor cortex dynamics is the prevalence of rotations. To quantify this, we decomposed the matrix $D$ into its symmetric and skew-symmetric components $D = D_{sym} + D_{skew}$. If dynamics are dominated by rotations, $D$ will be naturally close to skew symmetric, such that $D_{skew} \approx D$. If so, the dynamical fit provided by $D_{skew}$ will be both high and nearly as good as the dynamical fit provided by $D$.

Motor cortex dynamics were dominated by rotations in a way that SMA dynamics were not. For monkey Ba, the fit provided by $D_{skew}$ was more than six-fold better for motor cortex versus SMA (Fig. 6a, compare middle blue and black bars; $R^2 = 0.74$ versus 0.11). For monkey Ax, the fit provided by $D_{skew}$ was almost three-fold better for motor cortex versus SMA (Fig. 6b, compare middle blue and black bars; $R^2 = 0.62$ versus 0.23). These differences were statistically significant (for the conservative test: $p < 0.001$ for monkey Ba and $p < 0.05$ for monkey Ax). The different

dominance of rotational dynamics is also apparent when comparing within each area. For SMA, the $R^2$ associated with $D_{skew}$ was at most half as large the $R^2$ associated with $D$ (Fig. 6a, b, compare middle and left black bars). For motor cortex the $R^2$ associated with $D_{skew}$ was almost as high as the $R^2$ associated with $D$ (Fig. 6a, b, compare middle and left blue bars). For the muscles, the $R^2$ associated with $D_{skew}$ was negative (Fig. 6a, b, middle magenta bar) and significantly different from that in motor cortex by both tests. (Negative values of $R^2$ are possible if $D$ is far from skew-symmetric).

In addition to considering $D_{skew}$ (the skew-symmetric component of the best-fit dynamics matrix), we also considered $D_{skew}^*$, the skew-symmetric matrix that provides the best fit. For SMA, the $R^2$ associated with $D_{skew}^*$ was low (Fig. 6a, b, right black bar). For motor cortex, the $R^2$ associated with $D_{skew}^*$ was higher (Fig. 6a, b, right blue bar) and only modestly less than the $R^2$ associated with $D$. Comparing SMA versus motor cortex, the $R^2$ associated with $D_{skew}^*$ was statistically different via bootstrap (for the conservative test: $p < 0.001$ for monkey Ba and $p = 0.01$ for monkey Ax). For the muscles, the $R^2$ associated with $D_{skew}^*$ was small and statistically different from that for motor cortex (for the conservative test: $p < 0.0001$ for both monkeys).

As suggested by inspection of Figs. 4 and 5, what little rotational structure was present in SMA occurred at frequencies lower than in motor cortex. To quantify frequency, we analyzed the purely rotational system described by $D_{skew}^*$. For SMA, all rotational frequencies were below 1.1 Hz for both monkeys and both dynamical planes (Fig. 6c, d, black bars). For motor cortex, rotational frequencies were higher: the greater was ~3 Hz and the lesser was between 1.5 and 2 Hz (Fig. 6c, d, blue bars). The 1.5–3

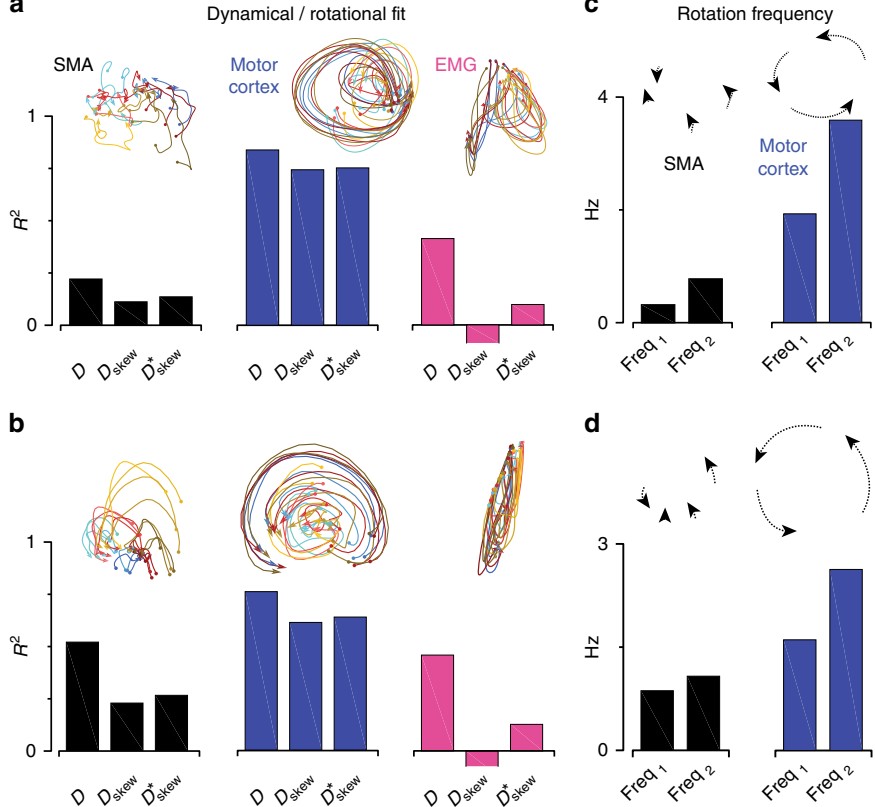

**Fig. 6** Quantification of dynamical structure. Insets indicate the nature of the structure being quantified. **a** Fit quality provided by unconstrained ($D$) and constrained ($D_{skew}$ and $D^*_{skew}$) linear dynamical fits. Fits are made over a 200 ms window, starting at movement onset, during which firing rates are changing rapidly. Results were robust to reasonable window sizes, so long as they included the time when responses were in rapid flux. For example, all p-values quoted in the text are lower (more significant) when using an earlier window: from 80 ms before movement onset until 150 ms after. Negative values, in the case of EMG, have been truncated. Data are for monkey Ba. **b** Same as **a** but for monkey Ax. **c** Rotational frequencies derived from $D^*_{skew}$ for SMA and motor cortex. Data are for monkey Ba. **d** Same as **c** but for monkey Ax

Hz frequencies in motor cortex are consistent with prior findings, and with models that use oscillatory dynamics to provide a basis-set for outgoing muscle-commands[6,41,42].

Thus, SMA and motor cortex differed in essentially every aspect of their dynamical structure. SMA population activity was less well fit by linear dynamics overall. SMA dynamics were not dominated by rotations, and what rotational structure was present occurred at frequencies lower than in motor cortex. A potential concern is that perhaps SMA and motor cortex are truly similar, but dynamics in SMA were missed because lower firing rates yielded lower signal-to-noise. This is unlikely for three reasons. First, dimensionality reduction effectively denoises responses by leveraging commonalities across neurons[68]. Given that we recorded 141 (monkey Ba) and 186 (monkey Ax) neurons from SMA, signal-to-noise is unlikely to be a large issue. Second, if signal-to-noise were poor in our SMA recordings, the variance captured by HDR should be much lower for SMA. In fact, both HDR and PCA captured slightly more variance for SMA versus motor cortex (Supplementary Fig. 3). Third, condition-invariant structure would be weakened if noise dominated signal. Yet as will be described below, HDR readily identified condition-invariant structure for both SMA and motor cortex.

**Qualitative assessment of condition-invariant structure**. HDR revealed a condition-invariant shift in state for both areas. The third column of Figs. 4 and 5 shows the projection onto the first dynamical dimension and the first condition-invariant dimension. For motor cortex, the resulting two-dimensional view can be

thought of as taking the first dynamical plane and spinning it so as to view the rotations "on edge", revealing new structure in a third dimension. This view reveals a large condition-invariant translation: trajectories start at the left for every condition, and translate a roughly equal distance to the right. Following the translation, rotations (viewed on edge) occur on the right side of the plane. This structure agrees with prior findings in motor cortex and in simulated networks[5,42].

We found that SMA also displayed a condition-invariant translation of the neural state, something not previously reported. This condition-invariant translation occurred alongside selectivity for condition in other dimensions (e.g., the first dynamical dimension separated trajectories across conditions). Both SMA and motor cortex also showed a second dimension with condition-invariant structure (see subsequent analyses).

**Quantification of condition-invariant structure**. We assessed the degree to which projections onto the condition-invariant dimensions were truly condition invariant. To quantify "condition invariance", we divided the variance of the across-condition mean by the total variance across all times and conditions. If a signal is identical across conditions, condition invariance will be 100%. Conversely, a signal that varies strongly with condition can have condition invariance approaching 0%. Both SMA and motor cortex contained relatively pure condition-invariant structure (Fig. 7a, b). For SMA, condition invariance was 96% and 95% (monkey Ba and Ax). For motor cortex, condition-invariance was

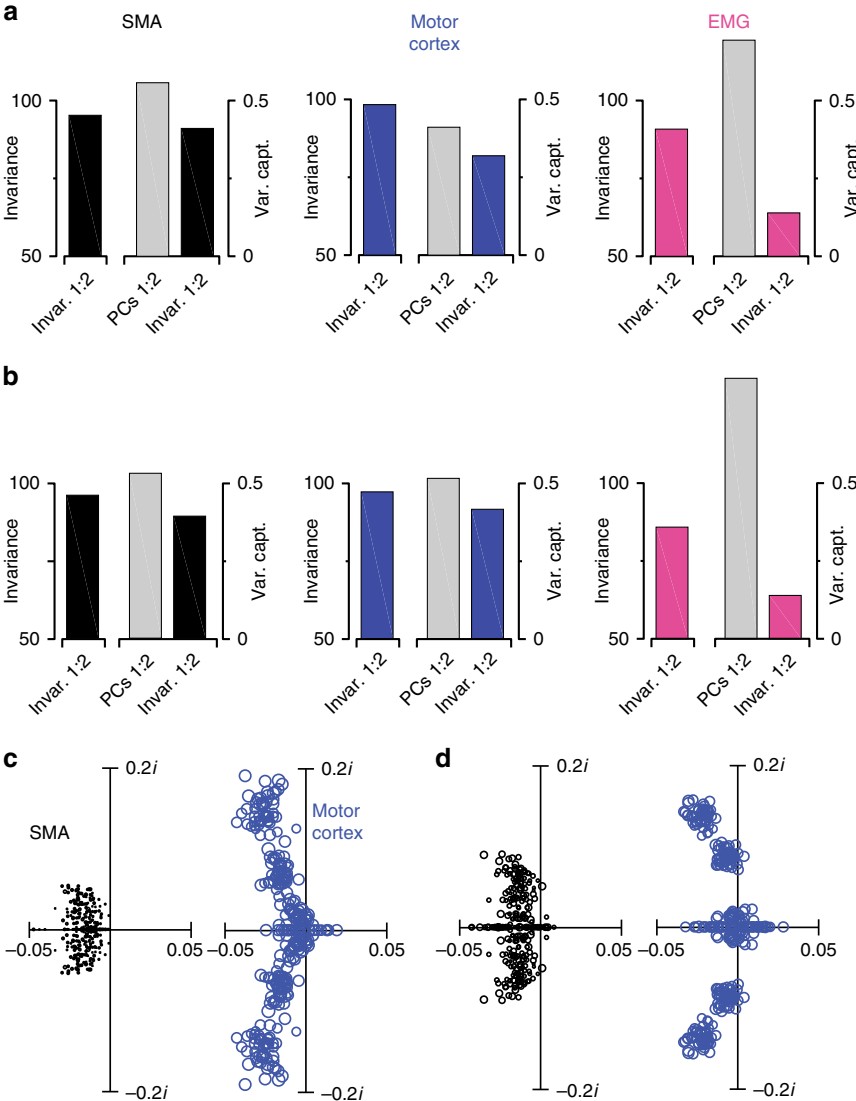

**Fig. 7** Quantification of HDR-based condition-invariant structure and PCA-based dynamical structure. **a** Quantification of the structure captured by the condition-invariant dimensions. In each subpanel, the leftmost bar indicates the empirical condition invariance of activity in the condition-invariant dimensions. The right plot indicates the total data variance captured by the two condition-invariant dimensions and that captured by the first two PCs (gray bar). These analyses focus on the time from −100 ms before movement onset until 200 ms after movement onset. This is the epoch during which the signal in the condition-invariant dimensions was changing rapidly. Data are for monkey Ba. **b** Same as **a** but for monkey Ax. **c** Eigenvalue spectra when fitting linear dynamics to the population response projected onto the top six principal components (PCs), rather than using HDR. The analysis was conducted for 100 bootstrap repetitions, each time re-drawing a new population of neurons (with replacement) from the recorded population. Fitting linear dynamics resulted in a set of six eigenvalues per bootstrap repetition. For visualization, only the first 50 sets of eigenvalues are shown. Circle sizes are proportional to fit quality. Data are for monkey Ba. **d**. Same analysis for monkey Ax

97% and 98%. For the muscles, condition invariance was modestly lower: 86 and 91%.

For both SMA and motor cortex, the condition-invariant signal was the dominant signal in the data: the two condition-invariant dimensions captured nearly as much variance as the first two PCs (Fig. 7a, b). For the muscles, condition-invariant structure was much weaker; the condition-invariant dimensions captured far less variance than the first two PCs. Thus, a condition-invariant signal is prevalent only in SMA and motor cortex. Subsequent analyses will explore whether the structure of that signal is similar in both areas.

**HDR-independent quantification of dynamical structure.** By design, the $f_{\mathrm{invar}}(W)$ and $f_{\mathrm{dyn}}(W)$ terms compete—any structure

captured by condition-invariant dimensions cannot be captured by dynamical dimensions. Might this cause HDR to miss dynamical structure, perhaps exaggerating the difference between SMA and motor cortex? Empirically this was not the case. Reducing the weighting of $f_{\mathrm{invar}}(W)$ by a factor of ten had little effect: dynamical fit quality changed little and the same difference persisted between SMA and motor cortex.

We also applied jPCA, which is more aggressive in seeking rotational structure. This becomes advantageous if there is a concern that rotational structure might be missed. jPCA revealed differences between SMA and motor cortex similar to those revealed by HDR. For monkey Ba, jPCA yielded a dynamical fit with an $R^2$ of $0.12 \pm 0.02$ for SMA versus $0.59 \pm 0.05$ for motor cortex (SEMs via bootstrap resampling neurons). For monkey Ax,

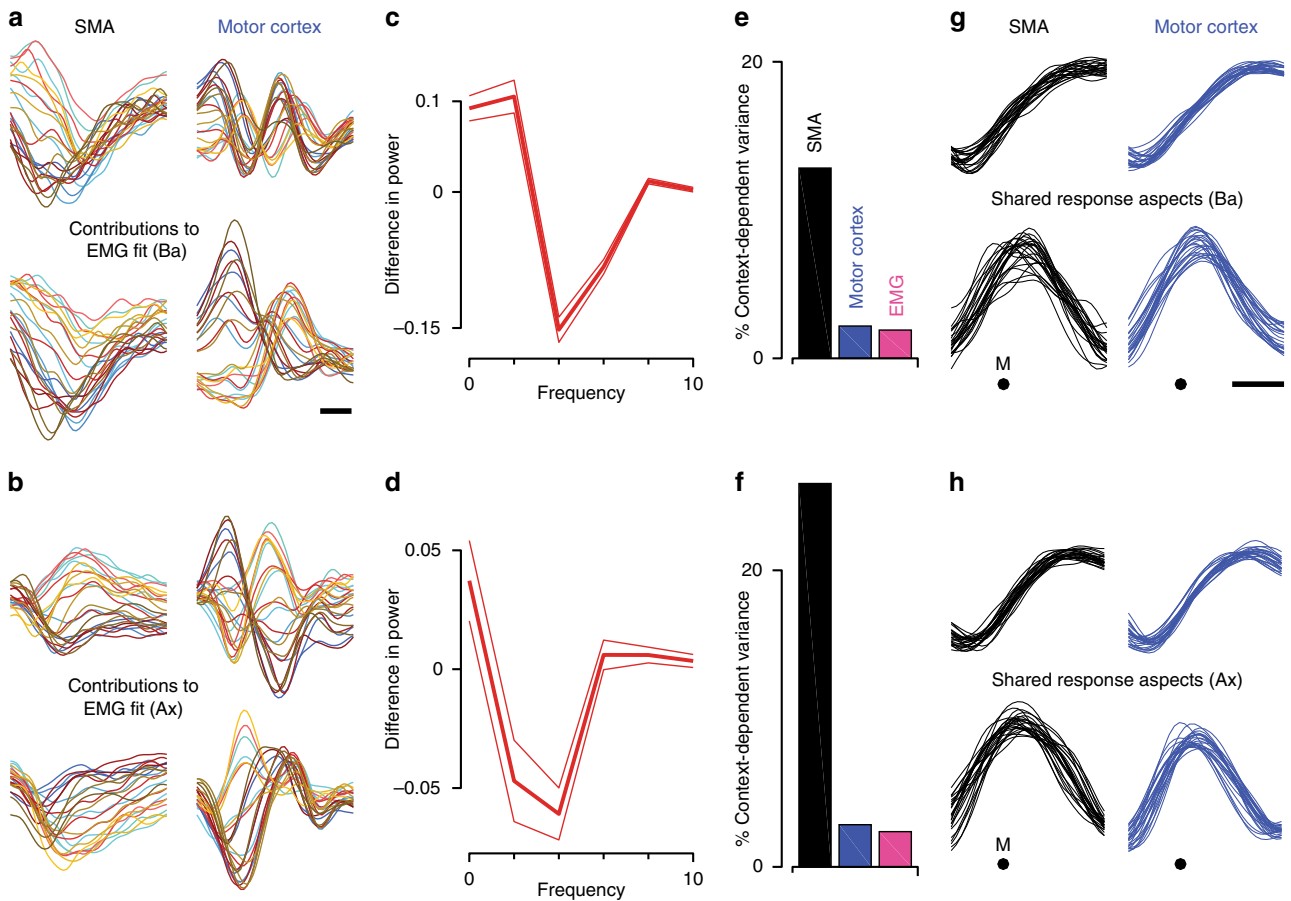

**Fig. 8** Potential contributions to outputs and internal computations. **a** Relative contributions of SMA and motor cortex when fitting muscle activity with neural activity. Analysis is based on the four dynamical dimensions for SMA, motor cortex, and the muscles. Activity in each EMG dimension was regressed against eight dimensions of neural activity, four each from SMA and motor cortex. The fit was the sum of contributions from SMA and motor cortex, shown separately in the left and right columns. Average $R^2$ was 0.58. For visualization, contributions are shown for two of the four muscle dimensions, chosen to highlight the greater contribution of motor cortex to multi-phasic structure. Each trace corresponds to one condition, plotted versus time over a 500 ms period starting 150 ms before movement onset (scale bar shows 100 ms). Data are for monkey Ba. **b** Same analysis for monkey Ax. The average $R^2$ was 0.72. **c** Quantification of the effects illustrated in panel **a**. Difference in normalized power, between SMA and motor cortex contributions, as a function of frequency. Positive values indicate greater power in SMA, negative values indicate greater power in motor cortex. Averages are taken across all conditions, and across all four dimensions of the EMG activity being fit. Flanking traces shown standard errors. **d** Same analysis for monkey Ax. **e** Percentage of total response variance due to activity varying with context. Data are for monkey Ba. **f** Same analysis for monkey Ax. **g** Response aspects that were most strongly shared between SMA and motor cortex, found via canonical correlation analysis. Dimensions were found within the space spanned by the six HDR dimensions, including the two condition-invariant dimensions and the four dynamical dimensions. Top and bottom rows plot the first and second canonical correlates. These are plotted versus time around the time of movement onset (M). Each trace corresponds to one condition. Scale bar shows 100 ms. Data are for monkey Ba. **h** Same analysis for monkey Ax

the corresponding $R^2$ was $0.28 \pm 0.05$ for SMA versus $0.55 \pm 0.04$ for motor cortex. As with HDR, rotational frequencies differed across areas. For monkey Ba, the highest rotational frequency was $0.65 \pm 0.13$ Hz for SMA versus $2.18 \pm 0.25$ Hz for motor cortex. For monkey Ax, the highest rotational frequency was $0.95 \pm 0.19$ Hz for SMA versus $2.19 \pm 0.32$ Hz for motor cortex.

Finally, we applied a less principled but very simple approach. We fit an unconstrained linear dynamical system to trajectories in the six dimensions, found via PCA, that captured the most condition-specific variance. Analysis was repeated, drawing new populations to allow bootstrap-based confidence intervals. Figure 7c, d plots the resulting eigenvalue spectra (each symbol plots one of the six eigenvalues for a given bootstrap repetition). Symbol size indicates fit quality, which was significantly lower for SMA versus motor cortex (monkey Ba: $R^2 = 0.15 \pm 0.02$ versus $0.63 \pm 0.05$; monkey Ax: $R^2 = 0.33 \pm 0.05$ versus $0.60 \pm 0.04$). Eigenvalue structure also differed. For SMA, the imaginary

(rotation-inducing) component never became as large as for motor cortex. For monkey Ba, the largest eigenvalue (of the non-bootstrapped data) had an imaginary component of $0.040i \pm 0.011i$ for SMA versus $0.144i \pm 0.17i$ for motor cortex ($0.144i$ corresponds to a frequency of 2.3 Hz). The corresponding values for monkey Ax were $0.044i \pm 0.017i$ versus $0.92i \pm 0.18i$. For motor cortex, eigenvalues tended to form two clusters at distinct frequencies. Little or no comparable structure was visible for SMA.

**Potential functions of SMA and motor cortex signals**. It has been hypothesized that rotational dynamics in motor cortex relate to the generation of multiphasic aspects of muscle activity within a movement[6,41,42,65] or across sub-movements[43]. To explore a possible connection with muscle responses, we regressed muscle activity versus SMA and motor cortex activity.

This analysis considered only the dynamical dimensions, which captured condition-specific response aspects. Each dimension of muscle activity was simultaneously regressed against all eight dimensions of neural activity (four each from SMA and motor cortex). The full fit (average $R^2$ of 0.58 and 0.72 for monkey Ba and Ax) was thus the sum of contributions from SMA and motor cortex (Fig. 8a, b). The SMA contribution was dominated by slower signals; higher frequencies were present primarily in the form of rapid ramps. In contrast, the motor cortex contribution contained overtly multiphasic structure (which differed across monkeys, as did the muscle responses being fit). To provide quantification, we computed the difference in the frequency content of SMA and motor cortex contributions (Fig. 8c, d). SMA showed a relative lack of contribution in the 3–4 Hz range for monkey Ba and the 2–4 Hz range for monkey Ax. Thus, the dynamical dimensions in SMA are less well-suited than those in motor cortex to contribute multi-phasic aspects of muscle activity.

Yet the dynamical dimensions in SMA contained a different type of structure: activity in the dynamical dimensions varied not only with direction, but also with the three contexts. These contexts varied regarding the rules and cues that determined when movement should be initiated. Yet the physical reaches were similar across contexts (Fig. 1a, b) as were both muscle activity and movement-related motor cortex activity[44]. The present analysis confirms prior findings: for motor cortex, nearly all variance in the dynamical dimensions occurred across time and/or reach direction. Very little (2–3%) occurred across contexts (Fig. 8e, f). The same was true of the muscles. However, we found that context had a much larger influence on the neural state in the SMA dynamical dimensions.

These findings are consistent with the oft-proposed role of SMA in movement initiation. A computation that determines when to move must consider the rules regarding when movement is allowed. However, other factors (e.g., visual cues and reward probability) also differed across contexts and could also be responsible for the observed effects. What is clear is that SMA responses co-vary with factors beyond the movements themselves, in a way that muscle activity and motor cortex activity do not.

**SMA and motor cortex share condition-invariant signals**. The above analyses reveal that the SMA and motor cortex population responses differ in multiple ways. To ask whether there also exist shared signals, we employed canonical correlation analysis (CCA). CCA returns projections of one dataset that are similar to (correlate with) projections of another dataset. We applied CCA to the six dimensions returned by HDR and examined the top two canonical variables; i.e., the most correlated projections. Each canonical variable is plotted versus time (Fig. 8g, h) to allow comparison of temporal structure. The top two canonical variables were highly correlated between motor cortex and SMA ($r = 0.99$ and 0.95 for monkey Ba; $r = 0.99$ and 0.96 for monkey Ax) and were very close to condition invariant. The condition invariance of the first canonical variable was 97 and 98% (SMA and motor cortex) for monkey Ba and 98 and 98% for monkey Ax. The condition invariance of the second canonical variable was 87 and 90% (SMA and motor cortex) for monkey Ba and 89 and 93% for monkey Ax.

The plots versus time in Fig. 8g, h are simply a complementary approach, relative to the state-space plots, of viewing condition-invariant structure. For example, the top row in Fig. 8g reveals a largely condition-invariant shift in state from low to high. Plotted in state-space this corresponds to a left-to-right shift in state common to all conditions, as in Fig. 4 (right column). Thus, both

SMA and motor cortex share a large condition-invariant signal that, for motor cortex, is known to be tightly linked to the timing of movement onset[5].

## Discussion

In highly interconnected networks, neural computations are believed to be instantiated by population-level dynamics[59,69,70]. We found that SMA and motor cortex differed in nearly every aspect of their dynamics. SMA was less-well described by linear dynamics. To the degree that dynamical structure was present in SMA it was not dominated by rotations. In particular, 1.5–3 Hz rotational structure was absent in SMA, despite being prevalent in motor cortex. These differences are notable because similar dynamics might have been expected for at least three reasons. First, single-neuron responses in SMA and motor cortex appear broadly similar during delayed-reach tasks, in both the present and prior studies[17,21,27,31]. Indeed, prior studies specifically sought clear differences between SMA and motor cortex during non-sequential reaches, but did not find them. Second, an appealing idea is that different areas apply a canonical computation to area-specific information[34,35]. Finally, it has remained controversial whether rotational dynamics in motor cortex reflect a specific computation, or are a generic property of complex, high-dimensional data[64]. The present results make clear that classes of dynamics can be specific to different areas.

SMA responses appeared disorganized in the dynamical dimensions. This is expected if responses do not obey the range of hypotheses embodied in the HDR cost function. Ideally, we would have employed other cost functions that embodied other hypotheses. However, neither past nor present results have yet yielded sufficiently concrete hypotheses regarding SMA to allow this strategy. That said, aspects of the present findings—in combination with prior work—suggest broad hypotheses that could be further refined. First, SMA possessed signals that could potentially contribute slower, non-multiphasic, features of muscle activity. Such a contribution is plausible given that SMA contributes to the corticospinal tract. Second, SMA activity reflected the behavioral constraints governing when movement should be initiated. Notably, such signals were present even during movement, after initiation. These observations suggest that the computations in SMA respect the broader context in which movement is executed.

Both SMA and motor cortex shared a condition-invariant shift in state. For both areas the condition-invariant shift was large, contributing almost as much variance as the first two PCs. Such a shift is not an inevitable consequence of surface-level response features[5] (e.g., overall increases in firing rate). For example, muscles activity often showed an overall increase across conditions, yet the muscle population exhibited a weak condition-invariant shift. We interpret the condition-invariant shift as reflecting neural events related to triggering movement. For example, in motor cortex, the timing of the condition-invariant shift is highly predictive of trial-by-trial variability in movement onset[5]. Furthermore, the condition-invariant shift is predicted by network models[5], where it does indeed serve the function of recruiting movement-generating dynamics.

The present data reveal that the shift is not only invariant with reach direction, it is also invariant with context. This is consistent with the tentative hypothesis that more "cognitive" computations (perhaps involving SMA) take context into account, but that the final signal that triggers movement onset no longer contains context information. That said, our data do not speak to the causal flow. The shift could arise in SMA and be immediately communicated to motor cortex, or the reverse could be true. It is also plausible that both areas inherit the shift from a common

input, or that the shift is generated by global dynamics in which many areas participate. These possibilities are difficult to resolve based on present observations. Synaptic latencies and transmission times occur on a timescale (a few milliseconds) an order of magnitude faster than the timescale on which shift unfolds (tens to hundreds of milliseconds), making it difficult to infer causality from chronology.

Our principal goal in applying HDR was to compare SMA and motor cortex. However, the use of HDR also clarifies the prevalence of rotational dynamics within motor cortex. The dynamical dimensions sought by HDR could have captured any type of linear dynamical structure: rotations, contractions, expansions, shear, or other forms of non-normal dynamics. In particular, expansions would be predicted if motor cortex signals encode a consistent preferred direction (e.g., if responses encode hand velocity or position). Instead, the dynamical dimensions for motor cortex were dominated by rotational structure, consistent with the emergence of rotational dynamics in network models trained to produce muscle activity[42]. In accord with this interpretation, dynamical dimensions for motor cortex contained features that matched, and thus may be contributing to, multiphasic aspects of muscle activity.

In summary, our results demonstrate that neural responses in SMA and motor cortex appear similar in many ways, including at the single-neuron level and when analyzed via standard methods. At the population level, both areas share a condition-invariant signal. Yet there are large differences that become apparent when focusing on dynamics. Furthermore, population activity in the two areas covaries differently with muscle activity and with task constraints. These findings argue that SMA and motor cortex are processing different types of information via different dynamics, presumably with different computational goals.

## Methods

**Subjects and task**. Subjects were two adult male macaque monkeys (*Macaca mulatta*) aged 10 and 14 years and weighing 11–13 kg. Daily fluid intake was regulated to maintain motivation to perform the task. All procedures were in accord with the US National Institutes of Health guidelines and were approved by the Columbia University Institutional Animal Care and Use Committee. Subjects sat in a primate chair facing an LCD display and performed reaches with their right arm while their left arm was comfortably restrained. Hand position was monitored using an infrared optical system (Polaris; Northern Digital) to track (~0.3 mm precision) a reflective bead temporarily affixed to the third and fourth digits.

We employed a center-out reaching task with three "contexts"[44]. Briefly, each trial began when the monkey touched and held a central touch-point. After the touch-point was held for 450–550 ms (randomized) a colored 10 mm diameter disc (the target) appeared in one of eight possible locations radially arranged around the touch point. In each context, similar reaches were made but the cue to initiate the reach was different. Touch-point and target color indicated context: red for cue-initiated, blue for self-initiated, and yellow for quasi-automatic. Target distance was 130 mm for cue and self-initiated contexts and began at 40 mm for the quasi-automatic context (final reach distance was similar in all contexts—see below). Trials for different contexts/directions were randomly interleaved.

In the cue-initiated context, after a variable delay period (0–1000 ms) the target suddenly grew in size (to 30 mm) and the central touch point simultaneously disappeared. These events served as the go-cue, instructing the monkey to make the movement. Reaches were successful if they were initiated within 500 ms of the go cue, had a duration <500 ms, and landed within an 18 mm radius window centered on the target. Juice was delivered if the monkey held the target, with minimal hand motion, for 200 ms (this criterion was shared across all three contexts).

In the self-initiated context, the target slowly and steadily grew in size, starting upon its appearance and ending when the reach began. Growth continued to a maximum size of 30 mm, which was achieved 1200 ms after target appearance (most reaches occurred before this time). The reward for a correct reach grew exponentially starting at 1 drop and achieved a maximum of 8 drops after 1200 ms. Monkeys were free to move as soon as the target appeared, but in practice nearly always waited some time: essentially all reaches occurred in the range from 600 to 1400 ms after target onset. In rare instances where no movement was detected within 1500 ms after target onset, the trial was aborted and flagged as an error.

In the quasi-automatic context, the target moved radially away from the central touch-point at 25 cm/s. Target motion began after a 0–1000 ms randomized delay period beginning at target onset. Target motion ended if a reach succeeded in bringing the hand to the target mid-flight. If the target was not intercepted (e.g., if

reach initiation was too slow) then the target continued moving until off the screen. Target speed and initial location (40 mm from the touch-point) were titrated, during training, such that the target was typically intercepted ~130 mm from the touch-point (the same location as the targets for the other two contexts). For successful interception, reaches had to land within an elliptical acceptance window (16 mm by 20 mm radius, with the long axis aligned with target motion). If the target was successfully intercepted, it grew in size to 30 mm and reward was delivered after the hold period.

In the present study, we analyze only trials where the delay period, for cue-initiated and quasi-automatic contexts, was >400 ms. Trials with shorter delays were included to encourage immediate and robust preparation, and because for the purposes of another study[44] we were interested in zero-delay trials in the quasi-automatic context. However, in the present study we wished to examine movement-period dynamics following the establishment of preparatory activity, in keeping with refs. [5–7,42].

**Neural and muscle recordings**. After subjects became proficient in the task, we performed sterile surgery to implant a head restraint. At the same time, we implanted a recording chamber centered over the arm area of motor cortex of the left hemisphere, including primary motor cortex (M1) and the dorsal premotor cortex (PMd). After recordings from M1/PMd were complete, the chamber was removed and a new chamber was implanted over the left-hemisphere SMA. Chamber positioning was guided by structural magnetic resonance images taken shortly before implantation. We used intracortical microstimulation to confirm that our recordings were from the forelimb region of motor cortex (biphasic pulses, cathodal leading, 250 μS pulse width delivered at 333 Hz for a total duration of 50 ms). Microstimulation of motor cortex typically evoked contractions of the shoulder and upper-arm muscles, at currents from 5 to 60 μA depending on the location and cortical layer. Microstimulation of SMA (total duration of 200 ms) sometimes caused full-arm movements reminiscent of a reach or an intentional arm movement at currents of ~20–100 μA. Other times, microstimulation of SMA up to 150 μA did not elicit any movement. As expected, thresholds were often higher in SMA. We thus used longer trains of microstimulation (and generally higher currents) in SMA simply because this was more effective in evoking movement, and we wished to verify that we were in arm-related SMA.

We recorded single-neuron responses using traditional tungsten electrodes (FHC) or one or more silicon linear-array electrodes (V-probes; Plexon) lowered into cortex using a motorized microdrive. For tungsten-electrode recordings, spikes were sorted online using a window discriminator (Blackrock Microsystems). For linear-array recordings, spikes were sorted offline (Plexon Offline Sorter). We recorded all well-isolated task-responsive neurons; no attempt was made to screen for neuronal selectivity for reach direction or any other response property. Spikes were smoothed with a Gaussian kernel with standard deviation of 20 ms and averaged across trials to produce peri-stimulus time histograms.

We recorded electromyogram (EMG) activity using intramuscular electrodes from the following muscles: lower and upper aspects of the *trapezius*, medial, lateral and anterior aspects of the *deltoid*, medial and outer aspects of the *biceps*, *brachialis*, *pectoralis*, and *latissimus dorsi*. The *triceps* were found to be minimally active (consistent with our prior observations in similar tasks) and were not recorded further. EMG signals were bandpass filtered (10–500 Hz), digitized at 1 kHz, rectified, smoothed with a Gaussian kernel with standard deviation of 20 ms, and averaged across trials to produce peri-stimulus time histograms.

**Population vector**. We first defined a preferred direction for each neuron by regressing movement-epoch firing rates (averaged over a 500 ms window centered on movement onset) versus horizontal and vertical target location. This was done for all 24 conditions (eight directions and three contexts). The population vector for a given condition was the sum of these direction vectors, each weighted by the average firing rate of the corresponding neuron.

**Preprocessing**. Prior to dimensionality reduction, each neuron's response was soft-normalized so that neurons with high firing rates had approximately unity firing-rate range (normalization factor = firing rate range + 5 spikes/s). This step follows our standard approach (e.g., refs. [5,6,10]), and ensures that the identified dimensions attempt to capture the response of all neurons, rather than a handful of high-firing-rate neurons. This is particularly important because many dimensionality reduction techniques (including PCA, HDR, and jPCA) focus on capturing variance. Without soft-normalization, a neuron with a firing rate of 75 spikes/s would contribute 25 times more variance than a neuron with a firing rate of 15 spikes/s.

**Hypothesis-guided dimensionality reduction**. Dimensionality reduction began by formatting neural (or muscle) responses as a matrix, $A$, where each column contains the responses of one neuron, concatenated across all times and conditions. $A$ is thus of size $CT{\times}N$ where $C$ is the number of conditions, $T$ is the number of time points and $N$ is the number of neurons. We also consider $A$, where each element describes the derivative of the firing rate for the corresponding condition, time and neuron. We seek a $CT{\times}K$ matrix, $X$, where each column describes one of $K$ latent variables ($K$ was set to six for all analyses). $X$ is found via projection:

$X = AW^T$, where $W$ is a $K \times N$ orthogonal matrix. Each latent variable is thus a weighted sum of individual-neuron responses, with the weights defining $K$ dimensions in neural space. Those weights were found by optimizing a cost function: $W = \text{argmin} f(W)$, with the constraint that $W$ is orthonormal. We used a tripartite cost function: $f(W) = f_{\text{rec}}(W) + f_{\text{invar}}(W) + f_{\text{dyn}}(W)$, with each term corresponding to one aspect of the guiding hypothesis.

The first term, $f_{\text{rec}}(W)$, is identical to the cost function used by PCA, and encourages the dimensions in $W$ to capture variance in $A$, such that $A$ can be reconstructed reasonably accurately from $X$. Because $W$ is an orthogonal matrix, $A_{\text{rec}} = XW = AW^TW$ is the optimal linear reconstruction of $A$ (in terms of minimizing mean squared error) from $X$. Thus, $f_{\text{rec}}(W) = \|A - A_{\text{rec}}\|_{\text{fro}}^2 = \|A - AW^TW\|_{\text{fro}}^2$. The squared Frobenius norm, $\|\cdot\|_{\text{fro}}^2$, is the sum of the squares of the individual elements. $f_{\text{rec}}(W)$ is small if $A$ can be reasonably well reconstructed from $X$. PCA can be thought of as a special case of HDR, where only the term $f_{\text{rec}}(W)$ is used. This corresponds to the 0th order hypothesis that the largest signals (those that most dominate the responses of single neurons) are important.

The second and third terms relate to the hypothesis that there exists condition-invariant structure in some dimensions and dynamical structure in other, orthogonal dimensions. We consider $W$ to be partitioned into two parts: $W = [W_{\text{invar}}; W_{\text{dyn}}]$. This results in a partitioned $X = \begin{bmatrix} X_{\text{invar}}, X_{\text{dyn}} \end{bmatrix} = A \begin{bmatrix} W_{\text{invar}}^T, W_{\text{dyn}}^T \end{bmatrix} = AW^T$. The second and third terms of the cost function relate to $X_{\text{invar}}$ and $X_{\text{dyn}}$, respectively.

The second term of the cost function, $f_{\text{invar}}(W)$, is small if $X_{\text{invar}} = AW_{\text{invar}}^T$ is invariant across conditions. We set $f_{\text{invar}}(W) = \text{trace}(W_{\text{invar}} C_{\text{across}} W_{\text{invar}}^T) / \text{trace}(W_{\text{invar}} C_{\text{ind}} W_{\text{invar}}^T)$. Where, $C_{\text{ind}}$ is the covariance matrix describing the aspects of $A$ that are condition-independent, and $C_{\text{across}}$ is the covariance matrix describing the aspects of $A$ that vary across conditions. $C_{\text{across}}$ is constructed by computing the covariance after removing, for each neuron, the cross-condition mean: the average firing rate at each time across all conditions. $C_{\text{ind}}$ is the covariance of the cross-condition mean itself. $f_{\text{invar}}(W)$, is thus small if, for the latent variables in $X_{\text{invar}}$, the cross-condition mean varies strongly with time but there is little variance among conditions around that mean.

The third term of the cost function, $f_{\text{dyn}}(W)$, attempts to identify a $W_{\text{dyn}}$ where the resulting $X_{\text{dyn}} = AW_{\text{dyn}}^T$ and its temporal derivative, $\dot{X}_{\text{dyn}} = \dot{A}W_{\text{dyn}}^T$ are linearly related, such that $\dot{X}_{\text{dyn}} \approx X_{\text{dyn}} D$ for some $D$. Assume that $D$ is chosen to provide the best fit, which can be accomplished by setting $D = X_{\text{dyn}}^\dagger \dot{X}_{\text{dyn}}$, where $\dagger$ indicates the pseudo-inverse. Then the variance accounted for by the fit is $\|X_{\text{dyn}}D\|_{\text{fro}}^2$. To find a $W$ that maximizes this variance, we set

$$f_{\text{dyn}}(W) = \|-X_{\text{dyn}}D\|_{\text{fro}}^2 = -\|X_{\text{dyn}}X_{\text{dyn}}^\dagger \dot{X}_{\text{dyn}}\|_{\text{fro}}^2 = -\|(AW_{\text{dyn}}^T)(AW_{\text{dyn}}^T)^\dagger(\dot{A}W_{\text{dyn}}^T)\|_{\text{fro}}^2$$

. $f_{\text{dyn}}(W)$ is thus small if the dimensions in $W_{\text{dyn}}^T$ capture structure whose temporal evolution (for all conditions) is well described by a linear dynamical system.

Minimizing $f(W)$ thus produces projections that balance capturing maximal data variance, finding a set of dimensions where trajectories are similar across conditions, and finding another set of dimensions where trajectories are fit by a linear dynamical system.

Iterative optimization is required to find the minimum-cost projection matrix $W$. Full details of the optimization technique can be found in ref. [67]. In brief, we wish to take gradient steps in the objective function $f(W)$ while respecting the constraint that $W$ is an orthogonal matrix ($W$ belongs to the Stiefel manifold). To do so, we first project the gradient $\nabla f(W)$ onto the tangent space of the constraint manifold, step in that direction, and then project the result back onto the constraint manifold. Although not guaranteed to reach the global optima (since the constraint manifold is nonconvex), this optimization is provably convergent to a local optimum. In practice we found the lack of global guarantee was not a major concern: for the datasets we analyzed, re-running optimization multiple times with different initializations resulted in final $W$ that spanned very similar spaces. For the datasets analyzed here, optimization converged relatively rapidly (~1 s on a 2017-era Apple Macbook Pro running Matlab 2016b).

**Finding the best-fit purely rotational linear dynamics**. For some analyses, we wished to ask how well trajectories in the dynamical dimensions could be described by purely rotational linear dynamics, if rotational dynamics were fit directly. To do so, we found $D_{\text{skew}}^* = \underset{D}{\text{argmin}} \|\dot{X}_{\text{dyn}} - X_{\text{dyn}}D\|_{\text{fro}}$, subject to the constraint that $D = -D^T$ (i.e., that $D$ is skew-symmetric). $\|\cdot\|_{\text{fro}}$ indicates the Frobenius norm, which in this case is simply the root-mean-squared error of the fit[6]. This is equivalent to performing regression, but with the constraint that the fit is provided by a purely rotational system. The eigenvalues of $D_{\text{skew}}^*$ are imaginary and were used to compute the rotational frequencies in Fig. 6c, d.

**Bootstrap tests for statistical significance**. Dimensionality reduction yielded latent variables with different properties for different datasets. For example, although HDR always sought dynamical dimensions where $\dot{X}_{\text{dyn}} \approx X_{\text{dyn}}D$, the goodness of this fit (the $R^2$) varied between SMA, motor cortex, and muscle populations. To ask whether $R^2$ differed between areas, one might be tempted to simply regress $\dot{X}_{\text{dyn}}$ against $X_{\text{dyn}}$ and compare the traditional confidence limits on $R^2$. However, this approach will overstate significance because, for both $X_{\text{dyn}}$ and $\dot{X}_{\text{dyn}}$, rows are not independent (e.g., nearby times tend to have similar states and similar derivatives). We therefore sought alternative approaches.

First, we employed a bootstrap in which we redrew, with replacement, 24 new conditions from the original 24. Each column of $X_{\text{dyn}}$ and $\dot{X}_{\text{dyn}}$ was modified to include data from the 24 redrawn conditions. We then recomputed the $R^2$. This process was repeated 1000 times to provide the sampling distribution. The $p$-value for a given comparison was the number of draws where the effect was not observed: e.g., if the $R^2$ for motor cortex was greater than the $R^2$ for SMA for both the original data and for 995/1000 bootstrap draws, then $p = 0.005$.

The bootstrap described above accounts for the possibility that the $R^2$ for one dataset might appear larger than that for another dataset due to "random" differences in individual-condition trajectories. This is reasonable, as the quality of the dynamical fit is in large part determined by whether different conditions obey the same dynamical flow-field. However, this approach does not address a different concern: perhaps the group of neurons recorded from one area simply happened to have more dynamical structure. This concern could be addressed by redrawing neurons, but that would not address the larger concern that different patches of cortex might be more or less "dynamical", e.g., perhaps motor cortex recordings simply happened to encounter a more dynamical group of neurons than did SMA recordings. To address this potential concern, we employed a very conservative bootstrap. This approach treated the four dynamical dimensions recorded from one area (e.g., motor cortex) and the four dynamical dimensions recorded from another area (e.g., SMA) as constituting eight dimensions in one larger undifferentiated "area". We then drew four random dimensions from this eight-dimensional space, and computed the $R^2$. This was done twice, and we computed the difference in $R^2$. We then collected a distribution of such differences across 1000 repetitions. This procedure asks how often one would observe a large difference in $R^2$ if there were truly no difference other than a random bias in which dimensions were sampled. The $p$-value was the number of such differences that were as large or larger than the empirical difference, e.g., if random differences were smaller than the empirical difference for 995/1000 repetitions, then $p = 0.005$.

**jPCA- and PCA-based approaches to assessing dynamics**. We applied the jPCA algorithm as described in ref. [6]. This involved three steps. First, for each neuron the cross-condition mean was removed such that the average firing rate (across conditions) was zero. Second, PCA was applied and the projection onto the top six PCs was retained. Third, we found the best-fit rotational linear dynamics (see above) that described the evolution of activity in those dimensions. Subsequent analysis focused on the features of these linear rotational dynamics, including fit quality and rotation frequency. The removal of the cross-condition mean (the first step) was important to ensure that PCA found dimensions where activity co-varies across conditions. Without this step, the first two dimensions are typically close to condition invariant. Notably, HDR did not employ this initial step because condition-invariant and condition-specific structure were isolated via a different (and generally preferable) method: by projection onto orthogonal dimensions. The PCA-based approach was nearly identical to the jPCA-based approach, but in the third step we fit with an unconstrained linear dynamical system. For both approaches, we applied a bootstrap procedure. The neural population was redrawn 100 times with replacement, and the analysis was repeated each time. This allowed us to put confidence intervals on measures such as the dynamical fit and the rotation frequencies, and to plot the eigenvalue spectra for multiple bootstrap repetitions to illustrate when values did or did not cluster.

**Code availability**. Code related to the dimensionality reduction approach of ref. [67] is provided at: https://github.com/cunni/ldr. Optimization code specific to the present study is available upon request to the corresponding author. The jPCA code package is available from the Churchland laboratory website: http://churchlandlab.neuroscience.columbia.edu

**Data availability**. All datafiles used to produce the figures and analyses in this manuscript are available, in matlab format, by direct request made to the corresponding author.

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

## Acknowledgements

We thank Yanina Pavlova and Sean Perkins for technical support. This work was supported by the Sloan Foundation, the Simons Foundation (SCGB#325233 and SCGB#542957), the Grossman Center for the Statistics of Mind, the McKnight Foundation, NINDS (1DP2NS083037), NIH CRCNS R01NS100066, NINDS 1U19NS104649, P30 EY019007, a Klingenstein-Simons Fellowship, the Searle Scholars Program, and an NIH Postdoctoral Fellowship (F32 NS092350).

## Author contributions

All authors collaborated in defining the general approach and key analyses. A.H.L. and M.M.C. designed the experiments. A.H.L. ran the experiments and recorded the data. J.P.C. implemented the optimization that allowed the central HDR analysis. A.H.L. and M.M.C. performed the analyses. All authors participated in interpreting results and writing the manuscript.

## Additional information

**Competing interests:** The authors declare no competing interests.

