## [Peer Review File · Nature Communications]

Reviewers' comments:

Reviewer #1 (Remarks to the Author):

This paper explores the differences in neural population dynamics between two areas in the motor pathway: SMA, and M1/PMd. Using a clever new dimensionality reduction technique termed hypothesis-guided dimensionality reduction (HDR), the authors observe that while there is a condition-invariant shift in neural state that's apparent with motor planning in both areas, M1/PMd exhibits strong linear rotational dynamics while SMA does not.

This is a well written paper that tackles an old question in motor control (what do different brain areas contribute to motor planning and execution?) in an entirely novel and interesting way. However, my enthusiasm for the work was substantially dampened by a lack of sufficient detail that accompanied many of their claims. Thus without a major restructuring of the paper I cannot recommend acceptance.

Major concerns:

1) The scope of the paper is too broad.

As it stands, the paper seems like a compromise between a methods paper that describes and validates a new dimensionality reduction technique, and a science paper that describes the differences in the dynamics between SMA and M1. This is too much to tackle in the Nature Communications format, and as a result it feels like neither task is accomplished sufficiently.

1a) On the methods front, HDR sounds like a really interesting and useful technique. I'd be excited to read a paper devoted solely to introducing and validating the tool. However, not enough information was presented for me to really understand the technique and be confident that subtle features of its implementation aren't skewing the results. For example, W is constrained to be orthonormal. Thus, latent variables that satisfy the f_{invar} constraint are going to influence the latent variables that satisfy the f_{dyn} constraint. In other words, there will be a trade-off between the three parts of the cost function such that if one part of the cost function is more dominant in a brain area, it may obscure findings for another part. In particular, imagine the scenario in which SMA is very dominated by the invar condition -- more so than M1. Might this obscure the ability to discover dynamic dimensions in SMA relative to M1? For these reasons, it's not clear that a failure to uncover dimensions that exhibit strong linear dependence means that strong linear dynamics are not present in the data. As another example, it's not clear to me how trial averaging might affect the data matrix A and any computations it performs. The data are aligned to two separate points: movement onset and target onset. How is the discontinuity handled? This seems important when evaluating derivatives and dynamics.

1b) On the science front, the two findings appear to be (1) that SMA and M1/PMd exhibit different dynamical structure, and (2) that this fact is not obvious using more traditional population analysis techniques. While I feel that (1) is sufficiently justified given the data presented, (2) is not. It's very hard to prove a negative. I understand the goal here really is to justify a population-level analysis, but if the authors want to stand by the statements in the manuscript on this point, they need to provide further support.

2) Certain statements require more quantitative support.

A number of statements in this paper were not backed up by appropriate quantification or statistical tests. Some examples:

2a) The statement "The population vector did not reveal any obvious differences between SMA and motor cortex." By eye, the plots kind of look similar, but they aren't even plotted in a superimposed fashion so it's hard to judge what to look for. I am aware that proving the PVA responses of these brain areas is the same is too much of a stretch. Nevertheless, I feel readers are going to interpret this statement as saying that the PVA cannot distinguish the responses in these two areas, which would be an overstatement. Either more the statement should be backed up rigorously, or it should be removed from the paper.

2b) The plots shown in Fig. 2 D, E, should include error bars.

2c) Page 11: "In the present case, the results of HDR reject the hypothesis that the SMA population response obeys approximately linear dynamics..." By what hypothesis test? What's the p-value? Further, the statement at the top of this paragraph ("The SMA population did not exhibit structure that could be well-described...") isn't backed up quantitatively until several pages later in the manuscript, which makes it hard to assess. All statements of this sort should be backed up with immediate quantification.

2d) Page 13-14: "The top two canonical variables were... very close to condition-invariant." Close relative to what? Later in the paragraph, the statement that the "... condition-invariant structure is both weaker and less fully shared with that in motor cortex" should be quantified.

2e) Page 14: Could the authors provide more explanation of the 'purity' computation?

Minor comments

1) Fig 1A: I don't see the star that's referred to in the caption.

2) Fig 1F: the color of the trace is referred to as magenta in the text and maroon in the caption.

3) I find the darker colors in Figure 2 hard to distinguish from one another. In particular, dark red looks much like dark yellow (both appear brown). Is there a way to adjust the color range to make them more differentiable?

4) Page 11: "It was thus much possible"

5) Page 11: "for nearly most models"

Reviewer #2 (Remarks to the Author):

This paper aims to compare the population dynamics of the supplementary motor area and the motor cortex. The authors apply a novel dimensionality reduction method to make the comparison between motor cortex and the SMA. In doing so, they show that while the motor cortex conforms to linear dynamics with a rotational component (which was already known), the SMA does not appear to conform as strongly to these linear-rotational dynamics.

The paper is very well written and work is technically well done, but there are some concerns about the novelty/relevance of the findings.

There are two elements of novelty in this work: 1) The distinction the authors make between

population dynamics in motor cortex and SMA and 2) the description of a novel hypothesis-driven dimensionality reduction method (HDR).

However, of these two, the description of the absence of rotational dynamics in SMA seems to add little to the current understanding of the role of the SMA in motor control or to our understanding of the functional and computational significance of rotational dynamics. Shortcomings in the experimental design (leading to the inability to study network dynamics during the preparatory period) also limit the study. Furthermore, apparent differences between the two monkeys with respect to the magnitude of rotational dynamics in SMA further complicate the interpretation of the results.

With respect to the second element of novelty, the development of the HDR method, it is not made experimentally clear why the use of the HDR is superior to the authors' previously used methods using jPCA (or combined dPCA and jPCA). It is not clear whether the distinction between SMA and motor cortex could have been made with a similar level of clarity using jPCA rather than HDR. Does jPCA reveal linear-rotational dynamics in SMA that HDR does not? If so, is this because of the inherent limitations in the jPCA method? If not, why is the much more time costly method of HDR a better method?

Further to this, the authors note that while one of the strengths of HDR is its agnosticism of the structure of the hypothesised linear dynamic (rotation, expansion etc.), this strength is also its weakness in that it is dependent upon the presence of a concrete hypothesis that can be reduced into a cost function. Because of this, while HDR could reveal difference that the SMA does not have the rotational dynamics that are present in motor cortex, the methods employed are unable to ascertain the actual structure of the population dynamics in SMA, nor are they able to indicate whether there is any dynamic structure at all in the SMA.

Overall, the work is technically well done, however, on conceptual ground, it does not appear to critically improve the current understanding of the computations underlying the execution of targeted movements or our understanding of the functional significance of rotational dynamics.

Concerning the technical innovation presented in this work, the HDR method, it is a potentially interesting development but it remains unclear how much of an advantage it provides over previously established methods.

Major points.

1) The standard population analyses are currently inadequate – statistical analyses and comparisons are needed. For example, page 7, 'The population vector did not reveal any obvious differences between SMA and motor cortex' is used to describe a lack of difference between brain regions using standard analyses.

Looking at figures 3 and supplementary figure 1, it appears in both animals that the vector lengths are lower in SMA than motor cortex for all conditions, and that the vector lengths are also more attenuated in SMA during the self-initiation context than in motor cortex. There also appears to be more variance in the SMA neuron activity between the times of target and movement onset (Figure 2A-B) than in the motor cortex – is this true? With regards to the PSTH, there also appears to be a longer period of activity in SMA than in motor cortex – this is present in both animals.

2) If the time-costly HDR analysis is to be used, we must first know that previous methods such as jPCA are inadequate to determine the differences in population dynamics between the motor cortex and SMA. Can jPCA provide us information that HDR does not provide us? This analysis should be done.

3) One of the main claims of the paper is the absence of rotational dynamics in SMA, however, there are some considerable differences between the two animals used that raise concerns as to the generalisability of the result:

Figures 4 ,5 and 7: Monkey Ba has a very disorganised structure in SMA while monkey Ax seems to have some rotational structure with a moderate fit ($R^2 = 0.52$)

Figure 6. The CCA analyses show quite different effects between animals - on the bottom row for the motor cortex vs EMG comparisons. Why is this? Some comment in the text is needed.

Minor points:

Figure 1C – these schematics seem a little vague. Can more precise locations of recording sites be given. Why are the colours of the recordings regions in a graded colour?

P12 – On the discussion of the signal-to-noise ratio in SMA. Another source of potential low signal-to-noise ratio is the apparent smaller population-vector lengths associated with SMA firing and preferred directions (figure 3 and sup fig 1).

P22- Much longer durations of microstimulations were used in SMA than in motor cortex. Moreover, some high current microstimulations did not produce movements. Why were these high amplitudes and durations used, and can some comment be provided on the apparent differences in the responses to stimulation of the two brain regions.

P25. It would be good to provide some idea of the differences in time-consumption between HDR and more standard methods.

Some typos:

P11 "It was thus much possible..."

P11 "for nearly most models..."

General Response and summary of key changes motivated by the reviewers' comments:

We very much thank the reviewers for their positive and supportive comments, and for useful and actionable criticism. We have performed new analyses and restructured the original manuscript accordingly. We believe the manuscript has benefitted greatly from this process.

Both reviewers noted the strengths of the study. For example, R1 stated that 'This is a well written paper that tackles an old question in motor control (what do different brain areas contribute to motor planning and execution?) in an entirely novel and interesting way.' R2 stated that 'The paper is very well written and the work is technically well done'. Additionally, the reviewers raised a number of very reasonable potential concerns.

One concern, shared by both R1 and R2, is that because HDR is a novel method, additional controls are needed to reassure the reader that HDR did not 'miss' structure in SMA. We agree: the reader needs to know that the difference between SMA and motor cortex isn't due to some peculiarity of a new method. R1 raises a legitimate potential concern regarding a 'tradeoff' that HDR makes between the condition-invariant and dynamical dimensions. R2 makes a related point: it is important to know whether older methods such as jPCA (which focus only on rotational dynamics and thus don't make the same tradeoff) also reveal a large difference between SMA and motor cortex. We have added a control to address the tradeoff concern. We have also re-analyzed the data using both a jPCA-based approach and an even-simpler PCA-based approach. The new analyses underscore that SMA and motor cortex have very different dynamical structure regardless of the analysis method chosen. Only motor cortex is well described by rotational dynamics, and what little rotational structure is present in SMA involves rotations at frequencies much lower than those in motor cortex. HDR is still the most principled approach. Yet we agree that it is still important to demonstrate that other approaches yield very similar results.

Both reviewers pointed out the need for more quantification for the population-vector and population-PSTH analyses. We have added that quantification.

Both reviewers expressed a desire for analyses that followed up on the differences in dynamics between SMA and motor cortex. Specifically, the reviewers desired analyses that inquired what those differences might mean in terms of potential function. We have thus added new analyses that explore the different functional contributions that SMA and motor cortex might make.

R2 noted that the difference between SMA and motor cortex was generally larger for monkey Ba versus monkey Ax. Specifically, the *unconstrained* (i.e., generic) HDR dynamical fit is quite different for monkey Ba and only moderately (though significantly) different for monkey Ax. The restructured manuscript and added analyses now better highlight that the largest differences pertain to rotational dynamics. This includes both the R^2 of the fit using rotational dynamics, and the nature of the rotational structure itself (e.g., the rotational frequencies). When it comes to these aspects, there is a large, robust, and statistically significant difference between SMA and motor cortex for both monkeys for every analyses we performed. This is true both for the original HDR analysis (e.g., the rotational fit is almost three-fold different for monkey Ax and over six-fold different for monkey Ba) and for the new jPCA and PCA-based analyses. Due to the additions and revisions, this is now much clearer than in the original manuscript (see also next point)

In response to comments by both reviewers, we restructured the flow of part of the results so that key quantification can follow more swiftly after showing the HDR-based projections. We have re-ordered a number of figures (and the relevant sections in the text) so that quantification of dynamics comes much sooner. This means that the many differences in dynamical structure are highlighted and quantified much earlier.

R1 noted a tension between documenting the properties of the method and delving into the science. We have restructured the manuscript to focus primarily on the science. We now lean more heavily on prior work that documents the basic ideas behind HDR, allowing the present study to focus on the scientific results. The newly-added jPCA- and PCA-based analyses (requested by both reviewers) also help to keep the focus on the science rather than a particular method.

We have made multiple other changes and additions as requested.

Comments, responses and quotes from the revised manuscript are in different fonts.

Responses, Reviewer 1

Reviewer's Remarks to the Author:

This paper explores the differences in neural population dynamics between two areas in the motor pathway: SMA, and M1/PMd. Using a clever new dimensionality reduction technique termed hypothesis-guided dimensionality reduction (HDR), the authors observe that while there is a condition-invariant shift in neural state that's apparent with motor planning in both areas, M1/PMd exhibits strong linear rotational dynamics while SMA does not.

This is a well written paper that tackles an old question in motor control (what do different brain areas contribute to motor planning and execution?) in an entirely novel and interesting way.

However, my enthusiasm for the work was substantially dampened by a lack of sufficient detail that accompanied many of their claims. Thus without a major restructuring of the paper I cannot recommend acceptance.

We have restructured the manuscript per the reviewer's comments (see below) and believe it has benefitted greatly. We thank the reviewer for the additional suggestions below. We have followed essentially all of these suggestions.

Major concerns:

1) The scope of the paper is too broad.

As it stands, the paper seems like a compromise between a methods paper that describes and validates a new dimensionality reduction technique, and a science paper that describes the differences in the dynamics between SMA and M1. This is too much to tackle in the Nature Communications format, and as a result it feels like neither task is accomplished sufficiently.

We have worked hard to mitigate the concerns expressed (also see responses to 1a and 1b below). This is first and foremost a scientific study, and we have revised the manuscript accordingly. We have made the following changes:

First, we now highlight that our methodology follows the framework developed in Cunningham and Ghahramani (2015). The present approach is simply an instance of the larger class they described. The present instance is defined by the cost function used. Because this cost function is tightly tied to our scientific hypothesis, this keeps the manuscript focused where it should be: on hypotheses and scientific results. Below is a key revised section of the Results:

To overcome these shortcomings, we employ a hypothesis-guided dimensionality reduction (HDR) methodology. Recent work observes that most dimensionality reduction methods implicitly or explicitly employ a cost function⁷⁶, and that different cost functions embody different hypotheses. For example, PCA embodies the simple hypothesis that the most relevant signals are the largest signals, while dPCA⁷⁵ embodies the hypothesis that different dimensions contain activity that co-varies with different task parameters. Here we follow this lead and leverage the suggestion that 'future linear dimensionality reduction algorithms can be derived in a simpler and more principled fashion'⁷⁶. We adopt a cost function tailored to our specific hypothesis: there may exist a projection of the data that captures a large percentage of response variance, with some dimensions capturing condition-invariant structure and other dimensions capturing structure described by linear dynamics. (page 8-9)

Second, we agree that it is important that the study not depend entirely on the use of the HDR method. Even though HDR is the most principled approach, it is novel, and many readers will wish to know whether other reasonable approaches yield the same result. We have therefore reanalyzed the data using both jPCA and an even simpler PCA-based approach. Results are very consistent across all methods: dynamical structure is stronger in motor cortex and rotational structure is much stronger. These results are described in a new section: *HDR-independent quantification of dynamical structure* (page 15).

1a) On the methods front, HDR sounds like a really interesting and useful technique. I'd be excited to read a paper devoted solely to introducing and validating the tool. However, not enough information was presented for me to really understand the technique and be confident that subtle features of its implementation aren't skewing the results...

We agree: some important information regarding implementation was missing. We have revised the Results to leverage the prior study (Cunningham and Ghahramani, 2015) that introduced the general framework behind HDR (treating dimensionality reduction as the optimization of a cost function). Our specific application of HDR can now be understood in that context. We have also added a section to the Methods to give details that had been lacking (also see subsequent comment):

Iterative optimization is required to find the minimum-cost projection matrix W . Full details of the optimization technique can be found in⁷⁶. In brief, we wish to take gradient steps in the objective function $f(W)$ while respecting the constraint that W is an orthogonal matrix (W belongs to the Stiefel manifold). To do so, we first project the gradient $\nabla f(W)$ onto the tangent space of the constraint manifold, step in that direction, and then project the result back onto the constraint manifold. Though not guaranteed to reach the global optima (since the constraint manifold is nonconvex), this optimization is provably convergent to a local optimum. In practice we found this lack of global guarantee was not a major concern: for the datasets we analyzed, re-running optimization multiple times with different initializations resulted in final W that spanned very similar spaces. (page 26)

1a continued) ...For example, W is constrained to be orthonormal. Thus, latent variables that satisfy the f_{invar} constraint are going to influence the latent variables that satisfy the f_{dyn} constraint. In other words, there will be a trade-off between the three parts of the cost function such that if one part of the cost function is more dominant in a brain area, it may obscure findings for another part. In particular, imagine the scenario in which SMA is very dominated by the invar condition -- more so than M1. Might this obscure the ability to discover dynamic dimensions in SMA relative to M1?

This is an insightful concern. It is now addressed in two ways. First, to directly address it we re-ran HDR after down-weighting the f_{invar} constraint by a factor of ten. Results were essentially identical: the quality of the dynamical fit (and the difference between SMA and motor cortex) was virtually unchanged. This was true as we swept the weight from 0.1 to unity. This largely rules out the possibility that the condition-invariant dimensions are 'stealing' structure from the dynamical dimensions and hurting the dynamical fit. This is now described in the Results:

We wished to ensure that the difference in dynamics between SMA and motor cortex was robust to the dimensionality reduction approach. We first considered that, by design, the $f_{\text{invar}}(W)$ and $f_{\text{dyn}}(W)$ terms compete – any structure captured by condition-invariant dimensions cannot be captured by dynamical dimensions. Might this somehow cause HDR to miss dynamical structure, perhaps exaggerating the difference between SMA and motor cortex? Empirically this was not the case. Even if we reduced the weighting of $f_{\text{invar}}(W)$ by a factor of ten, allowing the $f_{\text{dyn}}(W)$ term to dominate, results changed very little. In particular, the quality of the dynamical fits changed little and the same difference persisted between SMA and motor cortex for both monkeys (not shown). (page 15)

Second, we now employ non-HDR approaches (jPCA and PCA-based) that isolate dynamic from condition-invariant structure in a different way (the cross-condition mean is subtracted as a pre-processing step). Both jPCA and the PCA-based approach revealed much better dynamical fits (with a much clearer and higher-frequency rotational component) for motor cortex versus SMA. These results are shown in Figure 7C,D and described in the new section: *HDR-independent quantification of dynamical structure*.

2) Certain statements require more quantitative support.

We agree with both this general comment and the specific instances below. See responses to each comment below.

A number of statements in this paper were not backed up by appropriate quantification or statistical tests. Some examples:

2a) The statement "The population vector did not reveal any obvious differences between SMA and motor cortex." By eye, the plots kind of look similar, but they aren't even plotted in a superimposed fashion so it's hard to judge what to look for. I am aware that proving the PVA responses of these brain areas is the same is too much of a stretch. Nevertheless, I feel readers are going to interpret this statement as saying that the PVA cannot distinguish the responses in these two areas, which would be an overstatement. Either more the statement should be backed up rigorously, or it should be removed from the paper.

We agree. It is true both that more quantification was needed and that some statements were too strong. Regarding the population-vector analysis, we have now added quantification that allows direct comparison between properties of the population vector for SMA and motor cortex. This helps underscore the general similarity of the population-vector across the two areas, while also noting some small differences:

We found only small differences in the population vector between the two areas. We first considered the magnitude of the population vector, relative to the average firing rates of the contributing neurons. Comparing SMA with motor cortex, the magnitude of the population vector was on average slightly larger for monkey Ba (by 5%, N.S.) and smaller for monkey Ax (24%, $p < 0.01$). The SMA population vector was also slightly, but not significantly, less-well aligned with the actual target direction. The generally similar behavior of the population vector for both areas is consistent with the finding that firing rates in both areas vary with reach direction^{37,38}. (page 6)

We have also added similar quantification for the population PSTHs. This new quantification underscores the similarity between the population PSTHs for the two areas, while again documenting some small differences (specifically, preparatory activity is modestly more prevalent in SMA).

Population PSTHs revealed some modest differences between SMA and motor cortex. For example, population PSTHs indicate that the ratio of preparatory tuning versus movement-period tuning was slightly higher for SMA. We confirmed that this effect was representative of single-neuron responses. For each neuron we computed tuning as the standard deviation of firing rate across conditions, and took the ratio of preparatory tuning (300 ms before movement onset) to movement tuning (at movement onset). The median ratio was higher for SMA versus motor cortex (0.79 versus 0.52 for monkey Ba; 0.81 versus 0.64 for monkey Ax; $p < 0.01$ for both via rank sum test). For one monkey (Ba) tuning also tended to remain high slightly longer for SMA. However, a slight tendency in the opposite direction was observed for monkey Ax. Despite these modest differences, population PSTHs were strongly correlated between the two areas. For monkey Ba, correlations were 0.92, 0.96 and 0.92 for the cue-initiated, self-initiated, and quasi-automatic contexts. For monkey Ax, these correlations were 0.94, 0.96 and 0.92. The lower bound of the 95% confidence intervals was > 0.90 for all comparisons. (page 7)

We have been careful in the revision not to state that there are no differences between SMA and motor cortex when analyzed via traditional methods. SMA and motor cortex generally look very similar when compared using the population vector and PSTH. Yet there are a few quantitative differences and these are now documented.

2b) The plots shown in Fig. 2 D, E, should include error bars.

Agreed. These have been added.

2c) Page 11: "In the present case, the results of HDR reject the hypothesis that the SMA population response obeys approximately linear dynamics..." By what hypothesis test? What's the p-value? Further, the statement at the top of this paragraph ("The SMA population did not exhibit structure that could be well-described...") isn't backed up quantitatively until several pages later in the manuscript, which makes it hard to assess. All statements of this sort should be backed up with immediate quantification.

We agree both that the statement in question was poorly phrased (it has been removed) and that it is important for the manuscript to move more swiftly to the analyses that focus on quantification. To that end, we have restructured the flow of the manuscript by changing the order of figures and analyses. This allows a more rapid and natural progression from the projections in Figure 4&5 to the quantification of dynamics and rotational dynamics in particular (which is now in Figure 6).

In the revision, we have been careful to phrase conclusions conservatively until quantification arrives (and to flag to the reader that quantification will arrive soon). For example:

The SMA population (**Figure 4, top row**) did not exhibit structure that followed a clear dynamical flow-field (quantification to follow). (page 11)

2d) Page 13-14: "The top two canonical variables were... very close to condition-invariant." Close relative to what?

We agree this should be quantified. We now report the empirical 'condition-invariance' (the proportion of variance due to condition-invariant structure). It is very high:

The top two canonical variables were highly correlated between motor cortex and SMA ($r = 0.99$ and 0.95 for monkey Ba; $r = 0.99$ and 0.96 for monkey Ax) and were very close to condition-invariant. The condition-invariance of the first canonical variable was 97% (SMA) and 98% (motor cortex) for monkey Ba and 98% (SMA) and 98% (motor cortex) for monkey Ax. The condition-invariance of the second canonical variable was 87% (SMA) and 90% (motor cortex) for monkey Ba and 89% (SMA) and 93% (motor cortex) for monkey Ax. (page 18)

(Note that this analysis has been moved to Figure 8.)

Later in the paragraph, the statement that the "... condition-invariant structure is both weaker and less fully shared with that in motor cortex" should be quantified.

The analysis in question was the CCA-based comparison between EMG and motor cortex. This analysis has been removed to make room for a related but improved analysis relating neural and muscle activity (see new Figure 8A-D). Quantification (with error bars showing SEMs) for this new analysis is provided in panels C and D.

2e) Page 14: Could the authors provide more explanation of the 'purity' computation?

This is now better explained. Also, we now simply refer to it as 'condition-invariance'

For each population, we assessed the degree to which the projection onto the condition-invariant dimensions was truly condition-invariant. To quantify 'condition-invariance', we divided the variance of the across-condition mean by the total variance across all times and conditions. If a signal is identical across conditions, condition-invariance will be 100%. Conversely, a signal that varies strongly with condition can have condition-invariance approaching 0%. (page 14)

We have also changed the subsequent paragraph to quote percent condition-invariance, rather than percent of variance due to differences among conditions. While these metrics are equivalent, the former is more natural. We apologize that this was confusing in the original manuscript.

Minor comments

1) Fig 1A: I don't see the star that's referred to in the caption.

Fixed.

2) Fig 1F: the color of the trace is referred to as magenta in the text and maroon in the caption.

Fixed.

3) I find the darker colors in Figure 2 hard to distinguish from one another. In particular, dark red looks much like dark yellow (both appear brown). Is there a way to adjust the color range to make them more differentiable?

We have adjusted the brownish yellows and reds to be more different. This has improved things somewhat, but there are still 24 differently colored traces in one plot and we agree it can be challenging to

distinguish individual traces. We considered plotting each context separately, but this would triple the space required making it hard to show many examples. In the end we decided that the main point of the example PSTHs was simply to convey that responses are complex (and often multi-phasic) in both SMA and motor cortex. Fortunately this comes across clearly.

4) Page 11: "It was thus much possible"

Fixed.

5) Page 11: "for nearly most models"

Fixed.

Responses, Reviewer 2

Reviewer's Remarks to the Author:

This paper aims to compare the population dynamics of the supplementary motor area and the motor cortex. The authors apply a novel dimensionality reduction method to make the comparison between motor cortex and the SMA. In doing so, they show that while the motor cortex conforms to linear dynamics with a rotational component (which was already known), the SMA does not appear to conform as strongly to these linear-rotational dynamics.

The paper is very well written and work is technically well done, but there are some concerns about the novelty/relevance of the findings.

We thank the reviewer for their comments. We agree with the concerns expressed below and have modified the analyses and manuscript appropriately. The study has, we believe, benefitted greatly from these changes.

There are two elements of novelty in this work: 1) The distinction the authors make between population dynamics in motor cortex and SMA and 2) the description of a novel hypothesis-driven dimensionality reduction method (HDR).

However, of these two, the description of the absence of rotational dynamics in SMA seems to add little to the current understanding of the role of the SMA in motor control or to our understanding of the functional and computational significance of rotational dynamics...

We understand these points. Below, we first state what we believe *is* added by the HDR-based results (on their own). We then embrace the position advanced by the reviewer: more could and should be done to link the different dynamical structure to different potential functional roles. We have now done so with the addition of new analyses.

The central result of our study is that SMA and motor cortex have very different population-level dynamics. Historically it has been empirically unclear whether, for non-sequential reaches, SMA and motor cortex display similar response patterns (most results tended to suggest that they are similar). Our results reveal that, in terms of dynamics, SMA and motor cortex are different in multiple respects.

Yet the reviewer is quite correct: although different dynamics strongly suggest different computational contributions, the prior version of the manuscript did not go very far in exploring what those different contributions might be. We have therefore added a new set of analyses to the manuscript (Figure 8A-F) and an associated new section of the Results: SMA and motor cortex carry condition-specific signals with different potential functions.

These new analyses reveal different potential computational contributions of the dynamical dimensions in SMA versus motor cortex. Motor cortex, but not SMA, has activity patterns that could contribute to multi-phasic aspects of muscle activity. Conversely, SMA, but not motor cortex, has activity that covaries with the task demands that determine when and how movement should be initiated.

The new results are certainly not the last word regarding what computations SMA (or motor cortex) perform. Yet they add important context and make the connection (which had been missing in the original manuscript) between different population-level dynamics and potential differences in function. We are now able to ascertain that 1) dynamical structure is quite different between SMA and motor cortex, 2) aspects of those differences respect previously proposed functional roles of the two areas, and 3) both areas share a large signal previously proposed to be related to triggering movement initiation.

... Shortcomings in the experimental design (leading to the inability to study network dynamics during the preparatory period) also limit the study...

We agree that preparatory activity is an important topic and have revised the text to clarify why the present study focuses on movement-related activity. (We do consider preparatory activity in other studies – including studies under review and under preparation). A key section has been revised to convey these points:

In a separate study⁵⁰, we exploit these contexts to examine preparatory neural events in motor cortex alone. The evolution of preparatory activity in SMA will be the subject of future analysis and experiments. In the present study, we wish to compare movement-related dynamics, and ask whether a similar central motif is present in both areas.
(page 4)

...Furthermore, apparent differences between the two monkeys with respect to the magnitude of rotational dynamics in SMA further complicate the interpretation of the results.

This is an important point, and we describe below revisions to address it. These include new analyses, and a restructuring such that the critical quantitative comparisons and statistical tests come earlier. We are grateful to have been motivated to make these revisions; we 100% agree that it is important to demonstrate that, for both monkeys, differences in the prevalence of rotational dynamics are large, significant, and robust across analyses approaches. Fortunately, this is very much the case.

Most fundamentally, the difference in the rotational component of the dynamical fit is almost three-fold different for monkey Ax, and more than six-fold different for monkey Ba. Thus, the most critical comparison reveals a large and significant difference for both monkeys. The relevant section has been both revised and moved earlier to better highlight this difference:

Motor cortex dynamics were dominated by rotations in a way that SMA dynamics were not. For monkey Ba, the fit provided by D_{skew} was more than six-fold better for motor cortex versus SMA (**Figure 6A**, compare *middle blue* and *black bars*; $R^2 = 0.74$ versus 0.11). For monkey Ax, the fit provided by D_{skew} was almost three-fold better for motor cortex versus SMA (**Figure 6B**, compare *middle blue* and *black bars*; $R^2 = 0.62$ versus 0.23). These differences were statistically significant based on both bootstrap tests (for the more conservative test: $p < 0.001$ for monkey Ba and $p < 0.05$ for monkey Ax). The difference in the degree to which rotational dynamics dominate can also be appreciated by comparing within each area. For SMA, the R^2 associated with D_{skew} was at most half as large the R^2 associated with D (**Figure 6A,B**, compare *middle* and *left black bars*). For motor cortex the R^2 associated with D_{skew} was almost as high as the R^2 associated with D (**Figure 6A,B**, compare *middle* and *left blue bars*). (page 12)

The revised manuscript also better stresses that it is not only the goodness of fit that differs, but also the nature of the structure that is present. In particular, SMA has essentially no rotational component in the 1.5-3 Hz frequency range, while motor cortex has strong rotations in this range. This frequency range is important because it is relevant to the proposed function of rotational dynamics: producing multi-phasic aspects of muscle activity.

We realized that, in the original manuscript, quantification came much too late and key points were sometimes separated in different sections. The section *Quantification of dynamical structure*, now comes much earlier, and has been restructured. This section now documents that not only is the unconstrained linear dynamical fit better for motor cortex, that fit is dominated by rotations to a much greater degree. Furthermore, rotations are not only more dominant in motor cortex, but are also higher frequency.

We have also added a new analysis that assesses dynamical structure in a different way: by plotting the eigenvalues of an unconstrained linear dynamical fit to the projection onto the top six PCs (new Figure 7C,D). For both monkeys, the eigenvalues for motor cortex are consistent with a rotational dynamical system, with two semi-distinct frequencies. The eigenvalues for SMA do not have this structure, and the overall fit quality was significantly worse. This is described in the newly added section: *HDR-independent quantification of dynamical structure*.

At the reviewer's request, we have also added a comparison of SMA and motor cortex dynamical structure based on our older jPCA method. Again, differences are large and statistically significant:

jPCA revealed differences between SMA and motor cortex very similar to those revealed by HDR. For monkey Ba, jPCA yielded a dynamical fit with an R^2 of 0.12 ± 0.02 for SMA versus 0.59 ± 0.05 for motor cortex (SEMs via bootstrap resampling neurons). For monkey Ax, the corresponding R^2 was 0.28 ± 0.05 for SMA versus 0.55 ± 0.04 for motor cortex. As with HDR, there was a difference in the frequencies identified: for monkey Ba, the highest rotational frequency was 0.65 ± 0.13 Hz for SMA, versus 2.18 ± 0.25 Hz for motor cortex. For monkey Ax, the highest rotational frequency was 0.95 ± 0.19 Hz for SMA, versus 2.19 ± 0.32 Hz for motor cortex. (page 15)

Finally, the new analyses that relate to potential function (Figure 8A-F) further underscore that the differences between SMA and motor cortex are present across both monkeys and robust across a variety of analysis approaches.

In summary, the revised manuscript does a much-improved job of conveying and documenting the large differences in dynamics between SMA and motor cortex. It is true that monkey Ba tended to have larger effects for the majority of analyses, yet all analyses showed statistically significant differences between dynamics in SMA and motor cortex for both monkeys. For analyses that considered rotational structure (either fit quality or rotational frequency) the difference between SMA and motor cortex was consistently large. The newly added analyses regarding potential functional contributions reveal additional differences that were again present for both monkeys.

With respect to the second element of novelty, the development of the HDR method, it is not made experimentally clear why the use of the HDR is superior to the authors' previously used methods using jPCA (or combined dPCA and jPCA). It is not clear whether the distinction between SMA and motor cortex could have been made with a similar level of clarity using jPCA rather than HDR. Does jPCA reveal linear-rotational dynamics in SMA that HDR does not? If so, is this because of the inherent limitations in the jPCA method? If not, why is the much more time costly method of HDR a better method?

We have now added analyses based on jPCA and on PCA. These results are described in a new section: *HDR-independent quantification of dynamical structure*. jPCA results are described in the text and PCA-based results have been added to Figure 7 (panels C and D). All results strongly support the HDR-based findings: rotational structure is much weaker in SMA than in motor cortex (both in terms of the fit quality, and in terms of the rotational frequencies). Thus, while HDR is preferred (see below) it is certainly not necessary to see effects.

The revision also better justifies why we use HDR as the primary methodology. Briefly, HDR is more principled in a number of ways (e.g., optimization for variance explained and dynamical structure is performed jointly rather than sequentially) and also allows us to naturally separate condition-invariant and dynamical structure. HDR is also more conservative in the sense that it doesn't specifically seek rotational structure. Thus, HDR is the more principled and more conservative approach. This is described in the following revised text:

Although HDR takes additional effort to implement relative to existing methods, it is conceptually simple: it directly seeks structure predicted by a hypothesis expressed via a concrete cost function. Importantly, HDR optimizes jointly for all aspects of the hypothesized structure. In contrast jPCA employs PCA or dPCA and then seeks rotational structure^{11,12}, which could potentially cause structure to be missed. Unlike jPCA, the present approach does not focus on rotations *per se*, reducing concerns that the method imposes a particular form of dynamics. HDR is thus simultaneously more principled, more powerful, and more conservative than past approaches. That said, we stress that past approaches are not necessarily inadequate; indeed, key results will be replicated using jPCA and PCA. (page 10).

Further to this, the authors note that while one of the strengths of HDR is its agnosticism of the structure of the hypothesized linear dynamic (rotation, expansion etc.), this strength is also its weakness in that it is dependent upon the presence of a concrete hypothesis that can be reduced into a cost function. Because of this, while HDR could reveal difference that the SMA does not have the rotational dynamics that are present in motor cortex, the methods employed are

unable to ascertain the actual structure of the population dynamics in SMA, nor are they able to indicate whether there is any dynamic structure at all in the SMA.

This is a good point, and is now more explicitly recognized in the text.

We stress that this does not imply that the SMA population response is truly disorganized, simply that it is not well-described by the hypothesis of approximately linear dynamics. This highlights a key point of interpretation: the central virtue of HDR – employing a hypothesis-guided cost function – is both an advantage and a disadvantage. The absence of the hypothesized structure is revealing because it provides a test of the hypothesis. Yet in such cases, projections may not be particularly revealing. That said, SMA did exhibit other clear structure that could be identified by the HDR method, as will be illustrated in subsequent analyses. (page 11 of Results)

We also now better address this issue in the Discussion:

SMA responses appeared disorganized in the dynamical dimensions. This is expected if those responses do not obey the range of hypotheses embodied in the HDR cost function. Ideally, we would have employed other cost functions that embodied other hypotheses. However, neither past nor present results indicate sufficiently concrete hypotheses to allow the above strategy to be followed at this time. That said, aspects of the present findings – in combination with prior work – suggest broad hypotheses that could be further refined and tested... (page 19)

As an aside, we note that this limitation is not specific to HDR, but applies to all dimensionality reduction approaches (in one way or another they are all explicitly or implicitly based on cost functions). In general it is difficult to find revealing projections of high-dimensional data.

Overall, the work is technically well done, however, on conceptual ground, it does not appear to critically improve the current understanding of the computations underlying the execution of targeted movements or our understanding of the functional significance of rotational dynamics.

(This point is closely related to a point raised by the reviewer above, starting with ‘However, of these two...’ See our response above.)

We understand these points and have revised the manuscript accordingly. HDR reveals clear differences between SMA and motor cortex and reveals that the former largely lacks rotational dynamics (especially with higher frequencies). Given this, it is desirable to say something regarding what those differences may mean. One study can only do so much, but we agree that we should have done more. We have thus added a new set of analyses.

The new analyses are described in the section: *SMA and motor cortex carry condition-specific signals with different potential functions*. The findings are shown in the new Figure 8A-F. These new analyses address the potential functional significance of the differences between SMA and motor cortex. Briefly, in motor cortex (but not SMA) rotational dynamics produce firing rate patterns that could contribute to multiphasic features of muscle activity. Conversely, in SMA (but not motor cortex) activity in the dynamical dimensions lacks rotational structure, but contains a different type of information: activity covaries with the task demands that determine when and how movement should be initiated.

These results suggest different roles for each area (roles that are consistent with suggestions made from anatomical and lesion studies). These results don’t fully ‘nail down’ what either area does, but we still find them quite satisfying, and are glad that the reviewers suggested expanding the work in this direction.

Concerning the technical innovation presented in this work, the HDR method, it is a potentially interesting development but it remains unclear how much of an advantage it provides over previously established methods.

We have revised the text with this comment in mind. These revisions include the following:

Although HDR takes additional effort to implement relative to existing methods, it is conceptually simple: it directly seeks structure predicted by a hypothesis expressed via a concrete cost function. Importantly, HDR optimizes jointly for all aspects of the hypothesized structure. In contrast jPCA employs PCA or dPCA and then seeks rotational

structure^{11,12}, which could potentially cause structure to be missed. Unlike jPCA, the present approach does not focus on rotations per se, reducing concerns that the method imposes a particular form of dynamics. HDR is thus simultaneously more principled, more powerful, and more conservative than past approaches. That said, we stress that past approaches are not necessarily inadequate; indeed, key results will be replicated using jPCA and PCA. (page 10)

Furthermore, we have now added jPCA- and PCA-based analyses. Regarding dynamics, these analyses yield results very similar to HDR. However, unlike HDR, they cannot on their own segregate dynamical structure from condition-invariant structure. Doing so would require combining them with other methods. While we have done this in the past, it is a bit hacky, and we developed HDR because it solves the same problem in a more principled fashion.

The revised manuscript better conveys why we use HDR, and also that a set of reasonable approaches to assessing dynamics finds a consistent difference between SMA and motor cortex.

Major points.

1) The standard population analyses are currently inadequate – statistical analyses and comparisons are needed. For example, page 7, ‘The population vector did not reveal any obvious differences between SMA and motor cortex’ is used to describe a lack of difference between brain regions using standard analyses.

Looking at figures 3 and supplementary figure 1, it appears in both animals that the vector lengths are lower in SMA than motor cortex for all conditions, and that the vector lengths are also more attenuated in SMA during the self-initiation context than in motor cortex. There also appears to be more variance in the SMA neuron activity between the times of target and movement onset (Figure 2A-B) than in the motor cortex – is this true? With regards to the PSTH, there also appears to be a longer period of activity in SMA than in motor cortex – this is present in both animals.

We fully agree that more quantification was needed. This has now been added. Below is the revised text describing our findings for the population vector:

We found only small differences in the population vector between the two areas. We first considered the magnitude of the population vector, relative to the average firing rates of the contributing neurons. Comparing SMA with motor cortex, the magnitude of the population vector was on average slightly larger for monkey Ba (by 5%, N.S.) and smaller for monkey Ax (24%, $p < 0.01$). The SMA population vector was also slightly, but not significantly, less-well aligned with the actual target direction. The generally similar behavior of the population vector for both areas is consistent with the finding that firing rates in both areas vary with reach direction^{37,38}. (page 6)

Regarding the vector lengths, these are not directly comparable by inspection due to firing rate differences between the two areas (we apologize that we did not make this clear in the original manuscript). This is now explained in the figure legend:

Scaling is arbitrary and differs between the two brain areas, as they had different average firing rates (see main text).

For this reason, we now compare vector length directly (assessing its length relative to the average firing rate of the contributing neurons). The difference between SMA and motor cortex was small and inconsistent between the two monkeys (as noted in the revised text above).

The reviewer asked whether there is more variance in SMA during the delay period between target and movement onset. We have now analyzed whether variance across conditions (i.e., ‘tuning’) is greater in SMA during the delay period. As the reviewer anticipated, the ratio of delay-period tuning to movement-period tuning tended to be higher for SMA neurons. We also investigated whether tuning stays high for a longer period for SMA. This effect was small and not consistent between monkeys. We have revised the text to both note that there are some small but real differences between SMA and motor cortex that are revealed by the population PSTH, and also to quantify the overall similarity:

Population PSTHs revealed some modest differences between SMA and motor cortex. For example, population PSTHs indicate that the ratio of preparatory tuning versus movement-period tuning was slightly higher for SMA. We confirmed that this effect was representative of single-neuron responses. For each neuron we computed tuning as the standard deviation of firing rate across conditions, and took the ratio of preparatory tuning (300 ms before movement

onset) to movement tuning (at movement onset). The median ratio was higher for SMA versus motor cortex (0.79 versus 0.52 for monkey Ba; 0.81 versus 0.64 for monkey Ax; $p < 0.01$ for both via rank sum test). For one monkey (Ba) tuning also tended to remain high slightly longer for SMA. However, a slight tendency in the opposite direction was observed for monkey Ax. Despite these modest differences, population PSTHs were strongly correlated between the two areas. For monkey Ba, correlations were 0.92, 0.96 and 0.92 for the cue-initiated, self-initiated, and quasi-automatic contexts. For monkey Ax, these correlations were 0.94, 0.96 and 0.92. The lower bound of the 95% confidence intervals was > 0.90 for all comparisons. (page 7)

We also now address the fact that firing rates are more variable for SMA in the sense that they are 'noisier'. This is primarily due to lower firing rates in SMA.

2) If the time-costly HDR analysis is to be used, we must first know that previous methods such as jPCA are inadequate to determine the differences in population dynamics between the motor cortex and SMA. Can jPCA provide us information that HDR does not provide us? This analysis should be done.

We have now added jPCA-based analyses. See response to the above reviewer's comment that begins with, 'With respect to the second element of novelty...'

3) One of the main claims of the paper is the absence of rotational dynamics in SMA, however, there are some considerable differences between the two animals used that raise concerns as to the generalisability of the result:

Figures 4 ,5 and 7: Monkey Ba has a very disorganised structure in SMA while monkey Ax seems to have some rotational structure with a moderate fit ($R^2 = 0.52$)

(Note also response to the reviewer's comment above that begins with, 'Furthermore, apparent differences between the two monkeys...').

We apologize some points were not clear in the original manuscript. The restructuring of the revision has made things much clearer.

The R^2 for the *unconstrained* dynamical fit (not the rotational fit) is 0.52 for SMA for monkey Ax, significantly smaller than for motor cortex (0.76). For monkey Ba, this difference is larger (0.22 versus 0.84). Thus, the difference is present for both monkeys, but it is fair to say that it is very different for one monkey and only moderately different for the other.

The difference between SMA and motor cortex becomes larger, for both monkeys, when considering the rotational aspects of the dynamics. There the difference is almost three-fold for monkey Ax, and over six-fold for monkey Ba.

Not only did rotational dynamics provide a much poorer fit for SMA for both monkeys, what little rotational structure was present in SMA occurred at frequencies lower than those in motor cortex. This is important given the proposed role of rotational dynamics in motor cortex (producing high-frequency multiphasic features of muscle activity).

The original manuscript did not highlight the above points terribly well (much of the key quantification came too late). The manuscript has been heavily revised and the flow of figures and analyses has been changed. The section '*Quantification of dynamical structure*' now comes much earlier, and makes all of the key points quantitatively before summarizing the key points:

In summary, SMA and motor cortex differed in essentially every aspect of their dynamical structure. SMA population activity was less well fit by linear dynamics overall. SMA dynamics were not dominated by rotations, and what rotational structure was present occurred at frequencies lower than in motor cortex. (page 13)

The newly added jPCA-based and PCA-based analyses also reveal clear and consistent differences between SMA and motor cortex for both monkeys. We are pleased to have added these analyses and revisions, as we agree it is critical to show that effects are sizeable and significant for both monkeys

across a variety of comparisons. The revised manuscript does a much more compelling job of conveying that this is indeed the case.

Figure 6. The CCA analyses show quite different effects between animals - on the bottom row for the motor cortex vs EMG comparisons. Why is this? Some comment in the text is needed.

The reason was simple: the canonical correlates of motor cortex with EMG are different because the EMG patterns themselves are different for the two monkeys (this is common – even when velocity traces are similar). The relevant analysis has been replaced with a new one (Figure 8A,B). The reviewer's point still holds with regards to the new analysis, and we have added a comment as suggested.

In contrast, the contribution from motor cortex contained overtly multiphasic structure. (The particular multiphasic patterns were not the same for the two monkeys, which is expected because the patterns of muscle activity were also not the same.) (page 17)

Minor points:

Figure 1C – these schematics seem a little vague. Can more precise locations of recording sites be given. Why are the colours of the recordings regions in a graded colour?

We have added text to clarify the meaning of the shading:

C. Reconstructions of surface landmarks based on MRIs (see Supplementary Figure 1 for example MRI sections). Shaded regions in the reconstructions indicate where penetrations entered cortex, and are shaded darker to indicate where recordings were often made from deeper locations. (Figure 1 legend).

In response to this comment, we also added Supplementary Figure 1. This figure shows MRI sections, and provides additional information regarding recording locations (this is particularly helpful for SMA where most recordings were fairly deep, and thus on the medial wall rather than the surface).

P12 – On the discussion of the signal-to-noise ratio in SMA. Another source of potential low signal-to-noise ratio is the apparent smaller population-vector lengths associated with SMA firing and preferred directions (figure 3 and sup fig 1).

Population vector-lengths are indeed smaller for SMA in absolute terms (simply because firing rates were lower). However, they were not consistently smaller relative to those firing rates. This is now documented in the added quantification:

We first considered the magnitude of the population vector, relative to the average firing rates of the contributing neurons. Comparing SMA with motor cortex, the magnitude of the population vector was on average slightly larger for monkey Ba (by 5%, N.S.) and smaller for monkey Ax (24%, $p < 0.01$). (page 6)

P22- Much longer durations of microstimulations were used in SMA than in motor cortex. Moreover, some high current microstimulations did not produce movements. Why were these high amplitudes and durations used, and can some comment be provided on the apparent differences in the responses to stimulation of the two brain regions.

We now explain why we made this choice, and provide a relevant reference:

As expected, thresholds were higher in SMA⁸¹. We thus used longer trains of microstimulation (and generally higher currents) in SMA simply because this was more effective in evoking movement, and we wished to verify that we were in arm-related SMA. (page 24)

P25. It would be good to provide some idea of the differences in time-consumption between HDR and more standard methods.

Although at least one or two orders of magnitude slower than PCA, HDR converged quite rapidly for the datasets we analyzed. We have now added the following:

For the datasets analyzed here, optimization converged relatively rapidly (~1 second on an 2017-era Apple Macbook Pro running Matlab 2016b). (page 26)

Some typos:

P11 “It was thus much possible...” Fixed

P11 “for nearly most models...” Fixed

REVIEWERS' COMMENTS:

Reviewer #1 (Remarks to the Author):

The question of how different brain areas contribute to motor encoding has been of interest to neurophysiologists for decades. The approach taken here by Lara and colleagues yields unique insight into this problem: SMA and M1 neural populations express fundamentally different dynamical structure in their responses, implying different computations. The authors have addressed all of my previous concerns; this is a well written and well executed paper. I recommend acceptance.

Reviewer #2 (Remarks to the Author):

The authors have responded to our comments sincerely, competently and have addressed all our technical concerns.

The work is technically very well done but we remain unconvinced of the significance of the methodological and scientific advances put forward in this paper.

Regarding the methodological advances, the use of a hypothesis driven dimensionality reduction procedure (HDR) seems reasonable and provides a more principled way of finding dynamic structure than previous methods. However, when taken together with the additional analyses that have now been used in response to our observations (the use of jPCA and PCA to assess dynamic structure), it becomes clear that the differences in rotational structure between motor cortex and SMA can be found without the use of HDR. Indeed, the results of jPCA show a similarly striking lack of rotational fit to the SMA that was found using the newly developed HDR method. As such, we remain unconvinced of why one might opt for the more computationally and time costly methods of HDR when jPCA and PCA could have been used to show the same differences in dynamical structure between the SMA and motor cortex.

We do believe that there is potential in the new HDR, maybe the case study presented here simply does not do justice to it?

More importantly, our original concerns that this work contributes little to our understanding of the role of SMA in motor execution, unfortunately, also remain.

The additional analysis (Figure 8) that was implemented in response to our original comment does provide some information regarding the contextual effects on neural states in SMA that are not present in motor cortex, but these observations only provide indirect support for a commonly proposed role of SMA in non-motor aspects of movement and do not provide any further insight into the function of the SMA.

Specifically, it remains unclear why a general reader, with an interest in motor control, should care about the existence of less evident rotational dynamic states in SMA when compared to motor cortex. The authors have made little effort in clarifying why this matters.

As such, we still remain at a loss as to the role of SMA during the movement-related aspect of arm reaching and can only conclude that the SMA and motor cortex obey different population-level dynamics, but what purpose such dynamics serve remains untapped.

Albeit we do agree that not all can be answered in a single paper, given the target journal, the degree of scientific advancement provided by the work, both on the proposed role of rotational dynamics as well as on the role of the SMA in motor execution, might seem rather meagre to the general audience interested in motor control.

We thank the reviewers for their positive and supportive comments, and for useful and sincere criticism. We have made changes to the text in response to this latest round of comments.

Reviewer #1 (Remarks to the Author):

The question of how different brain areas contribute to motor encoding has been of interest to neurophysiologists for decades. The approach taken here by Lara and colleagues yields unique insight into this problem: SMA and M1 neural populations express fundamentally different dynamical structure in their responses, implying different computations. The authors have addressed all of my previous concerns; this is a well written and well executed paper. I recommend acceptance.

We thank the reviewer for their supportive comments and for many useful suggestions in the prior round. The manuscript improved greatly as a result.

Reviewer #2 (Remarks to the Author):

The authors have responded to our comments sincerely, competently and have addressed all our technical concerns.

We thank the reviewer for useful criticism that lead to improvements in the manuscript. As noted below, the reviewer suggests this work may be of limited interest to the motor control community. We respectfully disagree for reasons stated below. That said, we appreciate this point of view and have modified the text – especially the abstract – to highlight the aspects of our findings that we find novel and exciting.

The work is technically very well done but we remain unconvinced of the significance of the methodological and scientific advances put forward in this paper.

Regarding the methodological advances, the use of a hypothesis driven dimensionality reduction procedure (HDR) seems reasonable and provides a more principled way of finding dynamic structure than previous methods. However, when taken together with the additional analyses that have now been used in response to our observations (the use of jPCA and PCA to assess dynamic structure), it becomes clear that the differences in rotational structure between motor cortex and SMA can be found without the use of HDR. Indeed, the results of jPCA show a similarly striking lack of rotational fit to the SMA that was found using the newly developed HDR method. As such, we remain unconvinced of why one might opt for the more computationally and time costly methods of HDR when jPCA and PCA could have been used to show the same differences in dynamical structure between the SMA and motor cortex.

Certainly it is true that many of the results found via HDR can also be found via other methods (indeed it would be concerning if this were not so). That said, HDR is still a better – and much more extensible – method. Many of our scientific findings would be less compelling if found with other methods. As just one example, HDR sidesteps the oft-raised concern that jPCA finds rotational structure simply because it seeks such structure.

HDR is not particularly costly to run; for the present data, run-times were around one second. Other recently developed dimensionality reduction methods (e.g., GPFA) are far more time-consuming yet are still used by a broadening community.

We do believe that there is potential in the new HDR, maybe the case study presented here simply does not do justice to it?

HDR was actually quite important to the success of the present study. While many of our analyses could have been (approximately) accomplished with existing methods, following that path would have required using multiple methods in a non-optimal fashion. Results would have been less compelling because prior methods (including our own) are less formally principled, requiring additional controls to handle known shortcomings. Given our scientific goals, we of course wished to use the best available method.

We believe the text conveys the above points. As one example:

“We previously examined motor cortex responses using a method, jPCA, that seeks latent variables described by rotational dynamics. jPCA has two shortcomings given our present goals. First, when comparing areas, we wish to make fewer assumptions regarding the form of dynamics. Second, the central motif predicted by motor-cortex network models includes both rotational dynamics and a condition-invariant shift of the neural state⁴². We previously resorted to multiple methods (dPCA⁶⁶ followed by jPCA) to test for the presence of the central motif⁵. That approach is suboptimal; latent variables should ideally be found in a unified fashion.” (page 7)

We do agree that there is excellent future potential for HDR, as it provides a general recipe that can be adapted to different applications.

More importantly, our original concerns that this work contributes little to our understanding of the role of SMA in motor execution, unfortunately, also remain.

The additional analysis (Figure 8) that was implemented in response to our original comment does provide some information regarding the contextual effects on neural states in SMA that are not present in motor cortex, but these observations only provide indirect support for a commonly proposed role of SMA in non-motor aspects of movement and do not provide any further insight into the function of the SMA.

Specifically, it remains unclear why a general reader, with an interest in motor control, should care about the existence of less evident rotational dynamic states in SMA when compared to motor cortex. The authors have made little effort in clarifying why this matters.

We understand where the reviewer is coming from, yet disagree. To date, differential responses between SMA and motor cortex have been evident primarily during movement-sequence tasks. Multiple studies have sought clear differences during standard non-sequential reaching, yet have not found them. Our study is the first to reveal that SMA and motor cortex responses during non-sequential reaching are fundamentally different.

We concede that our results are revealing and intriguing only if one accepts that different classes of dynamics likely imply different computations. However, given recent theoretical and empirical results, that seems an increasingly safe assumption.

Regarding the functional role of rotational dynamics, in both present and prior work we have always attempted to be clear regarding our interpretation. We view rotational dynamics as likely sub-serving the generation of outgoing commands for multiphasic muscle activity (which may of course be further modified at the spinal level). We do agree that *some* aspects of our presents results are ‘negative’; we can say that SMA lacks rotational dynamics – and lacks approximately linear dynamics more generally – but we cannot say what dynamics (if any) SMA does obey. Still, we can say that SMA and motor cortex are unlikely to be performing redundant computations, even during non-sequential reaches. We can also say that SMA clearly contains much more ‘non-motor’ information regarding when and how movement should be initiated. As the reviewer says, a role in governing movement initiation based on task / contextual factors has indeed been proposed before, but the present results are (to our knowledge) the first demonstration that such factors have a large impact on SMA activity but not motor cortex or muscle activity.

As such, we still remain at a loss as to the role of SMA during the movement-related aspect of arm reaching and can only conclude that the SMA and motor cortex obey different population-level dynamics, but what purpose such dynamics serve remains untapped.

Albeit we do agree that not all can be answered in a single paper, given the target journal, the degree of scientific advancement provided by the work, both on the proposed role of rotational dynamics as well as on the role of the SMA in motor execution, might seem rather meagre to the general audience interested in motor control.

While we certainly find the contrasts between motor cortex and SMA to be exciting, we understand that others might legitimately have a different level of enthusiasm. We will say that, empirically, our results have not failed to garner interest when presented publicly. We thus believe – or certainly hope! – that the interest of the motor control community will not be as limited as the reviewer fears.

Indeed, given the rising appreciation that computations can be understood in terms of population dynamics, we believe the present results are quite timely. Without an examination of dynamics, many of the key differences (and some of the commonalities) between SMA and motor cortex population responses would remain largely invisible.